# Sex-differentiated placental methylation and gene expression regulation has implications for neonatal traits and adult diseases

Fasil Tekola-Ayele [1] ✉, Richard J. Biedrzycki[2], Tesfa Dejenie Habtewold[1], Prabhavi Wijesiriwardhana[1], Amber Burt [3], Carmen J. Marsit [3], Marion Ouidir [1,4] & Ronald Wapner[5]

Sex differences in physiological and disease traits are pervasive and begin during early development, but the genetic architecture of these differences is largely unknown. Here, we leverage the human placenta, a transient organ during pregnancy critical to fetal development, to investigate the impact of sex in the regulatory landscape of placental autosomal methylome and transcriptome, and its relevance to health and disease. We find that placental methylation and its genetic regulation are extensively impacted by fetal sex, whereas sex differences in placental gene expression and its genetic regulation are limited. We identify molecular processes and regulatory targets that are enriched in a sex-specific manner, and find enrichment of imprinted genes in sex-differentiated placental methylation, including female-biased methylation within the well-known *KCNQ1OT1/CDKN1C* imprinting cluster of genes expressed in a parent-of-origin dependent manner. We establish that several sex-differentiated genetic effects on placental methylation and gene expression colocalize with birthweight and adult disease genetic associations, facilitating mechanistic insights on early life origins of health and disease outcomes shaped by sex.

Sex differences in physiological traits and disease risk are pervasive, but the underlying mechanisms remain elusive. Phenotypic and developmental differences between males and females begin before birth, and differences manifest across the life span in disease prevalence, severity, and treatment response[1]. It has long been known that male fetus-bearing pregnancies have faster fetal growth rate[2], higher vulnerability to complications such as pre-eclampsia, placental abruption, fetal growth restriction, premature delivery, and postpartum hemorrhage, and greater risk of perinatal mortality[2–4]. Recent studies have shown that autosomal genetic factors contribute to sex-

differentiated phenotypes in adults, with some of these having prenatal molecular foundations[1,5]. Transcriptional regulation by methylation and gene expression can be a key underlying mechanism of these differences. Thus, investigating how fetal sex shapes the regulation of methylation and gene expression in tissues from early human development can unlock the molecular origins of health outcomes across the lifespan.

Successful fetal development relies on the placenta, a temporary organ during pregnancy at the maternal-fetal interface that exhibits sex differences in function, morphology, and response to

[1]Division of Population Health Research, Division of Intramural Research, Eunice Kennedy Shriver National Institute of Child Health and Human Development, National Institutes of Health, Bethesda, MD, USA. [2]Glotech, Inc., contractor for Division of Population Health Research, Division of Intramural Research, Eunice Kennedy Shriver National Institute of Child Health and Human Development, National Institutes of Health, Bethesda, MD, USA. [3]Gangarosa Department of Environmental Health, Rollins School of Public Health of Emory University, Atlanta, GA, USA. [4]University of Grenoble Alpes, Inserm, Team of Environmental Epidemiology applied to Reproduction and Respiratory Health, Institute for Advanced Biosciences, Grenoble, France. [5]Department of Obstetrics and Gynecology, Columbia University, New York, NY, USA. ✉e-mail: ayeleft@mail.nih.gov

environmental exposures[6-11]. Placental dysfunction can lead to pregnancy complications and early programming of diseases that manifest in later life in a sex-dependent fashion[12,13]. Accumulating evidence suggests that fetal sex-specific placental transcript expression[14-17] and methylation[18-20] may be among key molecular antecedents to male and female differences in placental function and neonatal outcomes[21,22]. Studies that integrated placental multi-omics have offered insights into the molecular bases of human complex traits[23,24]. Yet, to what extent sex influences the genetic regulation of both placental methylation and gene expression is unknown. Whether sex-differentiated placental methylation is linked with gene expression regulation in their nearby genomic region is unclear. Moreover, studies on sex differences in placental methylation and gene expression are rare in ancestrally diverse human populations.

In this study, we investigated the landscape of human placental methylation and gene expression impacted by fetal sex and its relevance to health and disease by integrating genome-wide placental methylation, gene expression, and genotype datasets from the *Eunice Kennedy Shriver* National Institute of Child Health and Human Development (NICHD) Fetal Growth Studies–Singleton cohort. We report (i) sex differences in placental methylation and its genetic regulation, and in placental gene expression and its genetic regulation; (ii) the biological significance of these sex-differentiated signals through functional annotations and omics integration; and (iii) the impact of these discoveries in advancing the molecular underpinnings of neonatal traits and adult diseases, revealing placental embedding of sex differences in human physiology and pathology.

## Results

### Overview of the study

The discovery analyses employed data from participants of the *Eunice Kennedy Shriver* National Institute of Child Health and Human Development (NICHD) Fetal Growth Studies–Singleton cohort[25]. There was no significant sex difference in maternal race/ethnicity, age, pre-pregnancy body mass index, and gestational age at delivery (Supplementary Data 1). Figure 1 provides a flow chart of the study. Briefly, a total of 301 samples (152 males and 149 females) were used to identify autosomal differentially methylated cytosine-phosphate-guanine sites (CpGs) between male and female placentas (sex-DM); a sub-sample of 291 (147 males and 144 females) was used to identify sex-biased methylation quantitative trait loci, which are genetic variants associated with different methylation levels in male and female placenta (sex-mQTL); 80 samples (43 males and 37 females) were used to identify differentially expressed autosomal genes between male and female placentas (sex-DE); and 71 samples (38 males and 33 females) were used to identify sex-biased expression quantitative trait loci, which are genetic variants associated with different gene expression levels in male and female placenta (sex-eQTL). Convergence between sex-biased methylation and gene expression was evaluated, and biological insights were gained through functional annotations. Phenotypic relevance was examined by testing the correlation of sex-DM and sex-DE with neonatal and placental anthropometry traits and through colocalization analysis of sex-mQTL and sex-eQTL with published genome-wide association studies (GWAS) loci for complex diseases and traits. The novelty of sex-DM and sex-DE was determined by

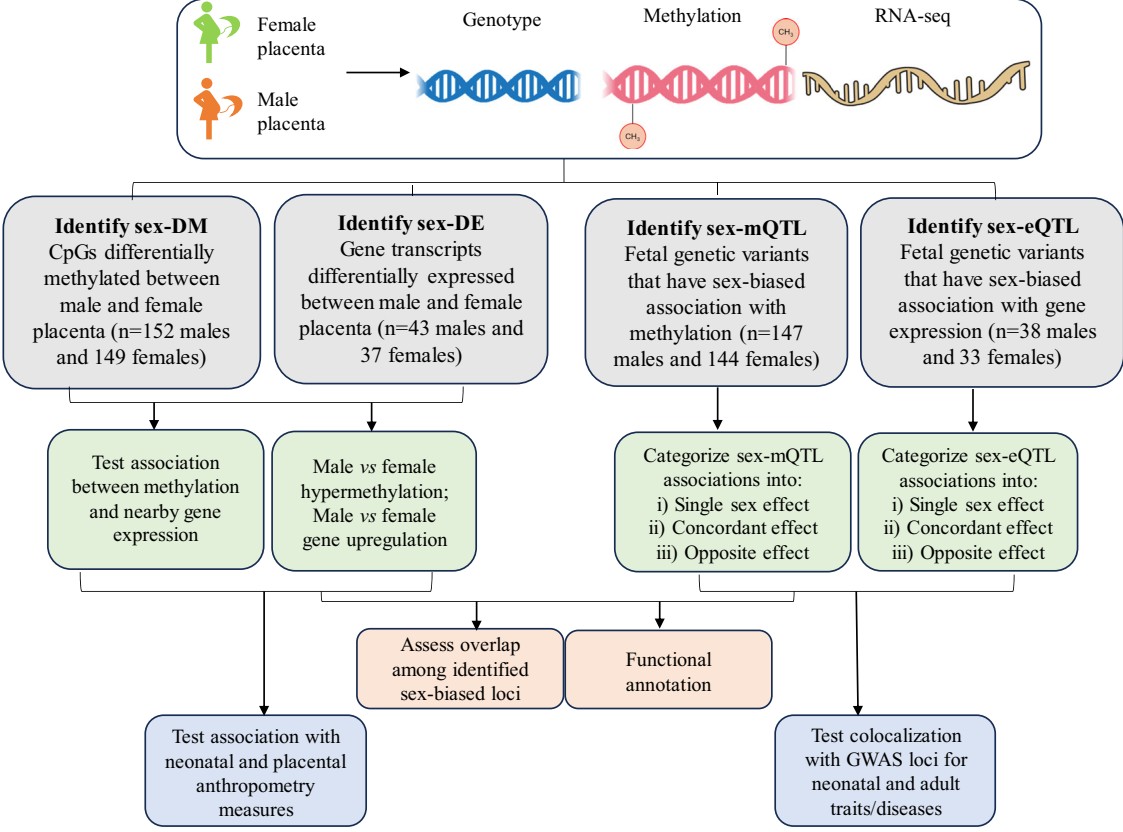

**Fig. 1 | Study flowchart.** The workflow summarizes the procedures in identifying sex-biased methylation, gene expression, and genetic regulation in the placenta; functional characterization through assessment of co-occurrence among identified loci and functional annotation using various databases; and illustration of the relevance of the identified loci in disease and health traits in neonates and adults. sex-DM: sex-differentially methylated cytosine-phosphate-guanine sites (CpGs); sex-DE: sex-differentially expressed genes; sex-mQTL: sex-biased methylation quantitative trait locus; sex-eQTL: sex-biased expression quantitative trait locus; GWAS: a genome-wide association study. Genotype (DNA), and RNA-seq (RNA) figures were generated by NIAID Visual & Medical Arts. (10/7/2024) and methylation was adopted. DNA. NIAID NIH BIOART Source. bioart.niaid.nih.gov/bioart/123. NIAID Visual & Medical Arts. (10/7/2024). RNA. NIAID NIH BIOART Source. bioart.niaid.nih.gov/bioart/452.

comparing with previous studies. Loci found to be sex-DE, sex-mQTL, or sex-eQTL were further assessed in an independent dataset from the Rhode Island Child Health Study (RICHS; 74 males and 74 females; more than 70% self-identified White; no sex difference in maternal race/ethnicity, age, pre-pregnancy body mass index, and gestational age at delivery; Supplementary Data 1)[26].

## Methylation differences between male and female placenta

We identified 6077 sex-DM at a false discovery rate (FDR)-adjusted $P < 0.05$. Out of the 6077 sex-DM, 41.1% (2497/6077) were previously unreported, and the remaining 58.9% have previously been reported with consistent effect direction[18–20,27–29] (Fig. 2a and Supplementary Data 2). Sex-DM CpGs were two times more likely to be male-hypermethylated than female-hypermethylated (66.9% vs. 33.1%), consistent with previous observations[18,19,28].

We assessed whether the sex-differentially methylated sites are correlated with nearby gene expression more than expected by chance. Methylation at sex-DM CpGs was more likely to be significantly correlated with nearby gene expression in the placenta at FDR-adjusted $P < 0.05$ than non-sex-DM CpGs on the methylation array (63/3636 vs. 1288/277343; $\chi^2 = 120.6$, $P = 4.67 \times 10^{-28}$). An additional 13 CpG-gene correlations were identified by testing the correlation of each sex-DM CpG against all genes within a 200 kb distance. These findings suggest that sex-dependent methylation is not randomly distributed across the genome but is clustered at CpG loci more likely to have impacts on gene transcript levels (Supplementary Data 3).

Differentially methylated sites may be involved in transcript regulation, but their precise effect is likely to depend on the genomic context[30]. In our data, positive sex-DM CpG-gene correlations (i.e., higher methylation correlated with higher gene expression) were more common than negative correlations. Moreover, the majority of CpGs positively correlated with gene expression were in the gene body, whereas the majority of those with negative correlation were in the gene transcription start site (TSS) (Supplementary Data 3). These findings are consistent with long-known theory and recent evidence that methylation within the gene body is often positively correlated with gene expression, while methylation in TSS is often negatively correlated with gene expression[30–36].

The top four strongest male-hypermethylated CpG-gene and top four strongest female-hypermethylated CpG-gene correlations are presented in Figs. 3a–d and 4a–d, respectively. Among male-hypermethylated sex-DM CpGs correlated with gene expression, methylation of cg11291313, located in *ZNF300* TSS, and *ZNF300* expression showed the strongest inverse correlation (Spearman r ($r_s$) = − 0.56, $P = 3.51 \times 10^{-7}$); and cg02563011, located within *CSMD1* gene body (167.4 kb from gene start), and *CSMD1* showed the strongest positive correlation ($r_s = 0.73$, $P = 3.44 \times 10^{-13}$). Previously, male bias has been found in a 225 kb locus of differentially methylated regions (DMRs) within the gene body of *CSMD1* and in transcript abundance of *CSMD1* in placenta[37], although the relationship of the DMRs with *CSMD1* transcript abundance was not elucidated. The two CpGs that we found to be correlated with *ZNF300* and *CSMD1* overlap with predicted active TSS-promoter and enhancer peaks, respectively, based on ENCODE and RoadMap Epigenomic annotations. Among female-hypermethylated sex-DM CpGs correlated with gene expression, cg01070760, located in *PSMA8* exon (193 bp from gene start), and *PSMA8* showed the strongest inverse correlation ($r_s = − 0.69$, $P = 3.07 \times 10^{-11}$); and cg25948255, located within the gene body of *CADM2* (504 kb from gene start), and *CADM2* showed the strongest positive correlation ($r_s = 0.65$, $P = 1.17 \times 10^{-9}$). The CpG linked to *PSMA8* overlaps with a predicted enhancer and with a long-range chromatin interaction loop (based on high throughput chromosome conformation capture,

Hi-C) in placental extravillous trophoblast (EVT) cells[38]. The CpG linked to *CADM2* overlaps with a predicted enhancer and active chromatin in EVT cells[38] and overlaps with a predicted polycomb-associated heterochromatin by ENCODE and RoadMap Epigenomic Annotations. *ZNF300*, *CSMD1*, and *CADM2* have been implicated in cancer cell proliferation, migration, and invasion[39–41], features mirrored by the invasive, immunologic, and angiogenic properties of placental cells[42]. *ZNF300* methylation has been correlated with placental morphology[43], and invasive placental trophoblast cells may drive sex differences at *ZNF300* methylation[27].

## Sex bias in genetic regulation of methylation in placenta

We identified 1839 sex-mQTL consisting of 1529 unique genetic variants and 521 unique CpGs (FDR-adjusted $P < 0.05$) (Fig. 2b and Supplementary Data 4). The mean (± standard deviation [sd]) distance between the genetic variant and target CpG in a sex-mQTL was 394.3 (± 306.4) kb pairs, suggesting that genetic variants anywhere within the 1 Mb distance can have sex-dependent effects on methylation in the placenta. As expected[44], the genetic variant's distance from its associated CpG was inversely correlated with its sex-biased effect on methylation ($r_s = −0.20$, $P = 7.32 \times 10^{-19}$). However, genetic variants may induce epigenetic regulation in the placenta through long-range chromatin interactions[38] and distal transcript regulatory elements such as enhancers[45,46]. Among our sex-mQTL SNPs, 37 SNPs overlap with predicted placental enhancers by Owen et al.[46], and 36 SNPs overlap with predicted placental enhancers by Zhang et al.[45]. The majority of SNPs that showed enhancer overlaps (i.e., 94.5% (35/37) in ref. 46 and 66.7% (24/36) in ref. 45) were more than 100 kb away from the sex-mQTL CpG (Supplementary Data 4).

A sex-QTL association can fall into a sex-specific, concordant, or opposite effect category based on the relative direction and magnitude of the allelic effect in males and females (see Methods section). In a sex-stratified analysis of the 1839 sex-mQTL pairs, 92.5% (1701/1839) had FDR-significant mQTL association in male and/or female placenta and further showed a significant difference in effect estimates between males and females at FDR-adjusted $P_{diff} < 0.05$. Among the 1701 sex-mQTLs with effects that differed by sex, 83.8% (1426/1701) were specific to one sex only, the strongest male-specific effect being for rs12242275/rs7093024-cg09082518 [*CCDC6*] ($\beta_{male(M)} = − 0.13$, $P_M = 6.77 \times 10^{-18}$, $\beta_{female(F)} = − 0.0026$, $P_F = 0.49$; $P_{diff} = 1.13 \times 10^{-13}$; minor allele frequency (MAF) = 5.2%) and the strongest female-specific effect being for rs4351362-cg14735364 [*NRF1*] ($\beta_M = − 0.003$, $P_M = 0.46$, $\beta_F = 0.12$, $P_F = 1.64 \times 10^{-12}$; $P_{diff} = 7.06 \times 10^{-10}$; MAF = 5.7%) (Figs. 5a–c). Downregulation of *CCDC6* in the placenta has previously been linked to preterm deliveries[47]. *NRF1* encodes a transcription factor involved in mitochondrial biogenesis and oxidative phosphorylation, and its reduced placental expression has been linked to preeclampsia[48]. 7.9% (134/1701) of sex-mQTLs had concordant effects in males and females (i.e., directionally consistent effects that differ in magnitude), rs530705354-cg14711243[*ACAD9*] being the top sex-differentiated with a stronger effect in males than in females ($\beta_M = 0.27$, $P_M = 1.31 \times 10^{-36}$, $\beta_F = 0.11$, $P_F = 2.45 \times 10^{-10}$; $P_{diff} = 9.12 \times 10^{-10}$; MAF = 6.2%). The remaining 8.3% (141/1,701) sex-mQTLs had opposite effects in males and females, with rs71304466-cg07784293[*WWTR1*] exhibiting FDR-adjusted association in both males and females and the strongest sex difference ($\beta_M = −0.04$, $P_M = 7.23 \times 10^{-8}$, $\beta_F = 0.01$, $P_F = 3.43 \times 10^{-3}$; $P_{diff} = 3.29 \times 10^{-8}$; MAF = 6%) (Supplementary Data 4). *WWTR1* encodes a transcription cofactor that regulates trophoblast cell fate during placentation, and defective expression in the placenta has associations with preterm birth, intrauterine growth restriction, and preeclampsia[49]. 778 sex-mQTL SNP-CpG pairs were available and tested in the RICHS dataset, and 19 were significant at FDR-adjusted

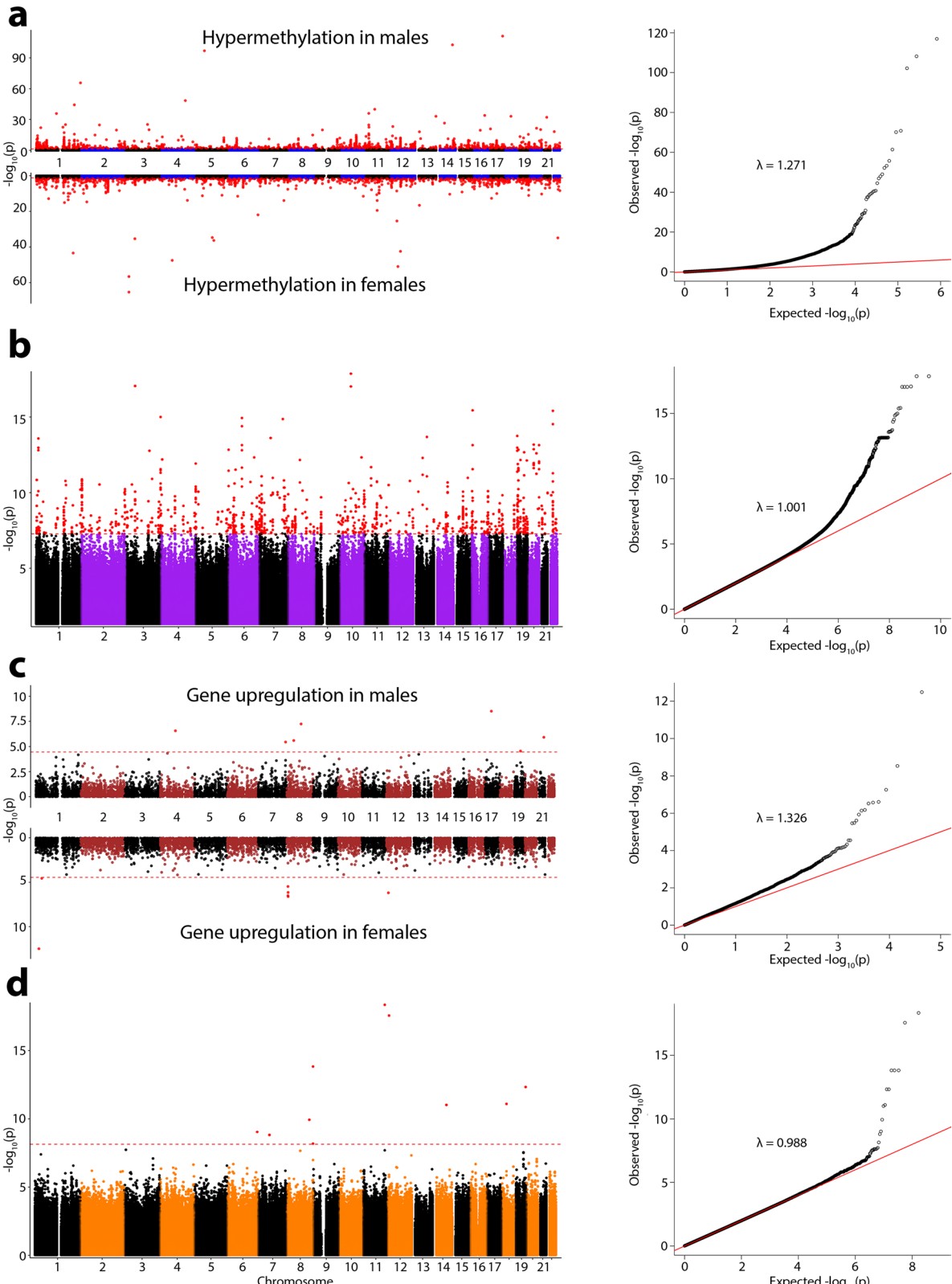

**Fig. 2 | Genome-wide placental sex-biased association results.** Each panel includes a Manhattan plot of -log10 *P*-values and the corresponding quantile-quantile plot of observed vs expected -log10 P-values and genomic control factor (λ). **a** sex-differences in methylation at cytosine-phosphate-guanine sites (sex-DM). **b** sex-biased genetic association with methylation (*cis* methylation quantitative trait loci (sex-mQTL). **c** sex-differences in gene expression (sex-DE). **d** sex-biased genetic association with gene expression (*cis* expression quantitative trait loci (sex-eQTL). Two-sided *t* tests were performed for (**a**) using limma with *n* = 301, df = 299, and

Benjamini-Hochberg adjustment and BACON correction applied for multiple testing corrections. Two-sided *t* tests were performed for (**b**) using MatrixEQTL with *n* = 291, df = 289, and Benjamini-Hochberg adjustment applied for multiple testing corrections. Two-sided likelihood ratio tests were performed for (**c**) using edgeR with *n* = 80, df = 1, and Benjamini-Hochberg adjustment applied for multiple testing corrections. Two-sided *t* tests were performed for (**d**) using MatrixEQTL with *n* = 71, df = 69, and Benjamini-Hochberg adjustment applied for multiple testing corrections. Red dots indicate statistically significant results.

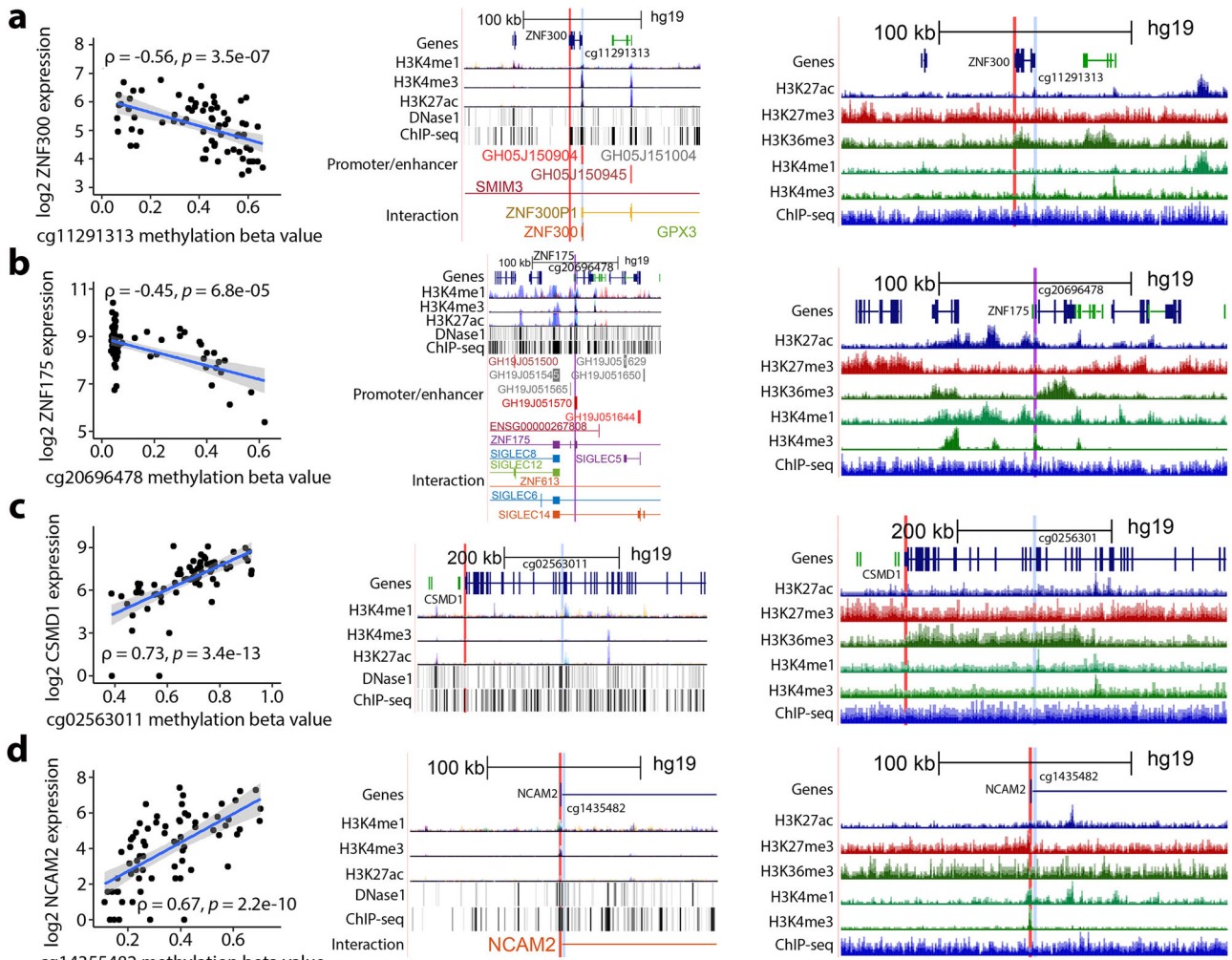

**Fig. 3 | Top four male-hypermethylated DNA methylation sites significantly correlated with nearby gene expression in placenta and overlap with regulatory regions.** Top associations were selected based on the strength of positive and negative correlations, two from each. The displayed figures include male-hypermethylated CpGs that showed the top two strongest negative correlations (**a**, **b**) and the top two strongest positive correlations (**c**, **d**) with their corresponding nearby genes and UCSC Genome Browser plots of regulatory sites. For each CpG, the left plot shows a correlation with gene expression in the placenta (two-sided Spearman correlation (ρ) with FDR-adjusted $P < 0.05$ based on S statistic was considered significant; $n = 71$); the middle plot shows UCSC Genome Browser's regulatory sites from ENCODE; the right plot shows UCSC Genome Browser's regulatory sites in placenta smooth chorion. Red highlights indicate the gene TSS; blue highlights indicate the CpG; and purple highlights indicate that the gene TSS and CpG are in close proximity to each other. Tracks include histone 3 lysine 4 monomethylation (H3K4me1: an enhancer mark), histone 3 lysine 4 trimethylation (H3K4me3: marks transcription start site), histone 3 lysine 27 trimethylation (H3K27me3: marks transcription repression), histone 3 lysine 36 trimethylation (H3K36me3: marks active transcription), and histone 3 lysine 27 acetylation (H3K27ac: marks active transcription) found in 7 cell lines from ENCODE and/or in placenta smooth chorion. Tracks labeled "DNase1" represent DNaseI hypersensitivity clusters in 125 cell types from ENCODE. Tracks labeled "ChIP-seq" represent ChIP-seq clusters in 338 factors and 130 cell types from ENCODE and ChIP-seq marks found in placenta smooth chorion. Tracks labeled "Promoter/enhancer" and "Interaction" represent GeneHancer Double Elite Regulatory Elements and GeneHancer Double Elite Clustered Interactions, respectively. Gray shade denotes a 95% confidence interval. Source data for Fig. 3a–d are provided as a Source Data file.

$P < 0.05$ and with consistent effect direction. The strongest male-specific mQTL association identified (i.e., rs12242275/rs7093024-cg09082518 [*CCDC6*]) was among those replicated in the RICHS dataset (Supplementary Data 5). One of our sex-mQTLs (rs34571066-cg13299927 [*ARHGEF10L*]) has previously been reported[20].

**Gene expression differences between male and female placenta**
We found 14 sex-DE (FDR-adjusted $P < 0.05$), of which sex-DE of *ANGPT2* has been previously reported[20] and 13 were previously unreported (Fig. 2c and Supplementary Data 6). In the RICHS dataset, 12 sex-DE transcripts were available, and 10 showed directionally consistent sex-DE associations, but none was significant after FDR adjustment (Supplementary Data 6). None of the 14 genes exhibited sex difference in non-placental tissues from the genotype tissue

expression (GTEx) portal[50], aligning with evidence of poor correlation of sex-differentiated autosomal gene expression between term placenta and 42 non-reproductive adult tissues from GTEx[51]. None of the sex-DE has placenta-specific RNA expression as compared to other human tissues based on tissue-specificity metrics in the Human Protein Altas (https://www.proteinatlas.org/about/download#protein_atlas_data) or in a placental RNA-sequencing study (Supplementary Data 6)[52,53].

**Sex bias in genetic regulation of gene expression in placenta**
We identified 13 previously unreported sex-eQTL (FDR-adjusted $P < 0.05$), consisting of 13 unique variants and 9 unique gene transcripts (Fig. 2d and Supplementary Data 7). The mean ($\pm$ sd) distance between a sex-eQTL SNP and target gene was 419.9 ($\pm$ 216.5) kb pairs,

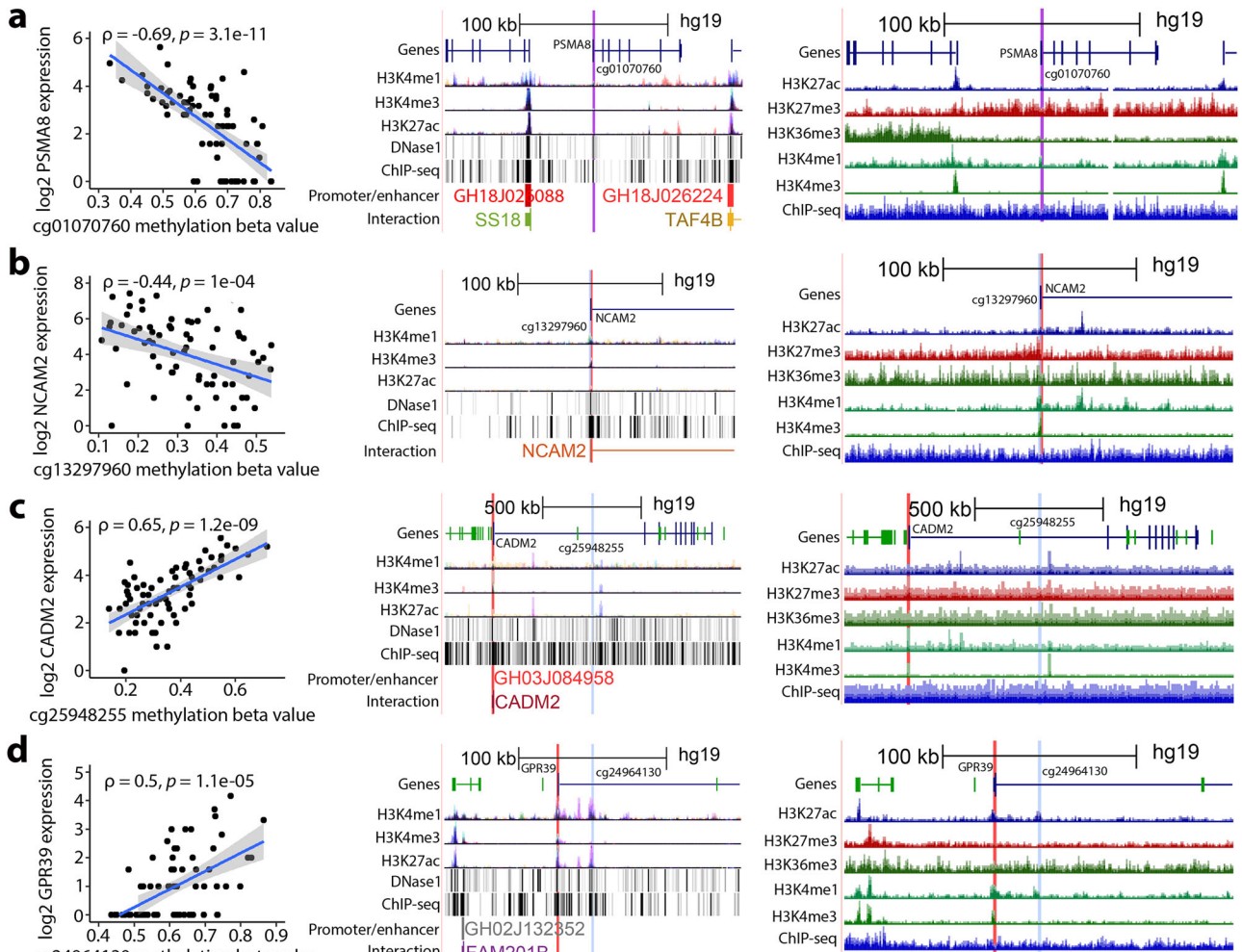

**Fig. 4 | Top four female-hypermethylated DNA methylation sites significantly correlated with nearby gene expression in placenta and overlap with regulatory regions.** Top associations were selected based on the strength of positive and negative correlations, two from each. The displayed figures include female-hypermethylated CpGs that showed the top two strongest negative correlations (**a**, **b**) and the top two strongest positive correlations (**c**, **d**) with their corresponding nearby genes and UCSC Genome Browser plots of regulatory sites. For each CpG, the left plot shows a correlation with gene expression in the placenta (two-sided Spearman correlation (ρ) with FDR-adjusted $P < 0.05$ based on S statistic was considered significant; $n = 71$); the middle plot shows UCSC Genome Browser's regulatory sites from ENCODE; the right plot shows UCSC Genome Browser's regulatory sites in placenta smooth chorion. Red highlights indicate the gene TSS; blue highlights indicate the CpG; and purple highlights indicate that the gene TSS

and CpG are in close proximity to each other. Tracks included histone 3 lysine 4 monomethylation (H3K4me1: an enhancer mark), histone 3 lysine 4 trimethylation (H3K4me3: marks transcription start site), histone 3 lysine 27 trimethylation (H3K27me3: marks transcription repression), histone 3 lysine 36 trimethylation (H3K36me3: marks active transcription), and histone 3 lysine 27 acetylation (H3K27ac: marks active transcription) found in 7 cell lines from ENCODE and/or in placenta smooth chorion. Tracks labeled "DNaseI" represent DNaseI hypersensitivity clusters in 125 cell types from ENCODE. Tracks labeled "ChIP-seq" represent ChIP-seq clusters in 338 factors and 130 cell types from ENCODE and ChIP-seq marks found in placenta smooth chorion. Tracks labeled "Promoter/enhancer" and "Interaction" represent GeneHancer Double Elite Regulatory Elements and Gene-Hancer Double Elite Clustered Interactions, respectively. Gray shade denotes a 95% confidence interval. Source data for Fig. 4a–d are provided as a Source Data file.

covering a broad region within the 1 Mb genomic window. In a sex-stratified analysis of the 13 sex-eQTL pairs, 92.3% (12/13) had FDR-significant eQTL association in male and/or female placenta and further showed a significant difference in effect estimate between males and females (FDR-adjusted $P_{diff} < 0.05$). All 12 sex-eQTLs exhibited a genetic effect in one sex only, the strongest female-specific effect being for rs79910893-*LPCAT3* ($\beta_M = -0.002$, $P_M = 0.96$, $\beta_F = 1.42$, $P_F = 4.72 \times 10^{-7}$; MAF = 5.6%) and the strongest male-specific effect being for rs11986287-*SLC52A2* ($\beta_M = 0.88$, $P_M = 1.26 \times 10^{-12}$, $\beta_F = -0.03$, $P_F = 0.14$; MAF = 5.6%) (Fig. 6a–c). None of the sex-eQTLs overlap with sex-eQTLs in 44 non-placental tissues from GTEx (https://gtexportal.org/home/downloads/adult-gtex/qtl)[50], consistent with evidence that sex differences in genetic regulation of gene expression are highly tissue-specific[50,54,55]. Out of the 13 sex-eQTLs, only rs10892219-*ATP5MG*

was available in the RICHS dataset and did not exhibit significant association after FDR adjustment.

## Overlap of sex-biased methylation and gene expression regulation

To identify loci which may concurrently exhibit sex differences in methylation level, gene expression level, genetic regulation of methylation, and genetic regulation of gene expression, we looked for overlaps among our results (i.e., sex-DM, sex-DE, sex-mQTL, sex-eQTL). Only two overlaps were found: (i) A co-occurrence of sex-DM and sex-DE: male-biased cg25364822 methylation at the transcription start site of *FNDC5* and female-biased expression of *FNDC5*. In turn, higher methylation at cg25364822 was significantly correlated with lower *FNDC5* expression in males ($r_s = -0.36$, $P = 0.023$), but not in

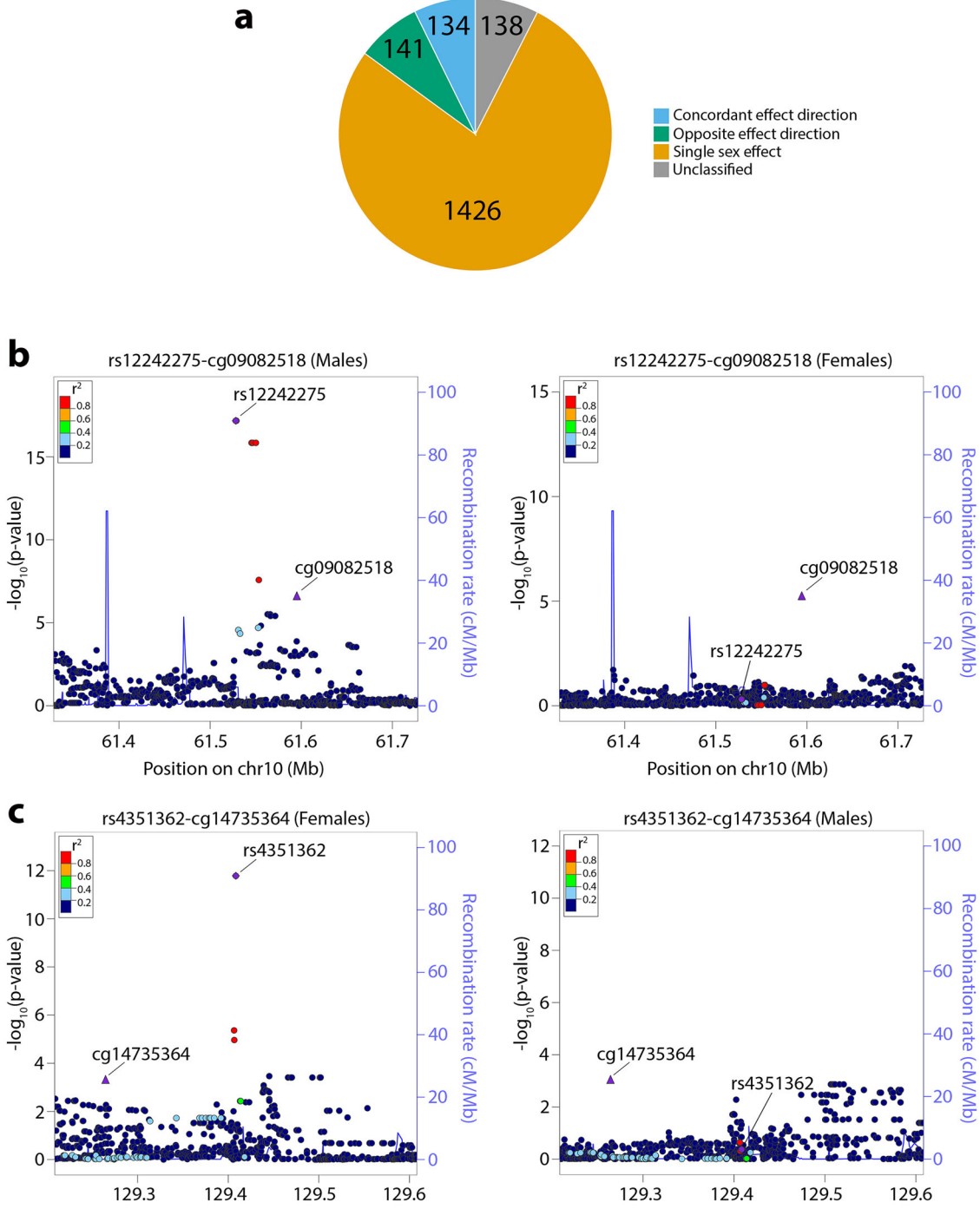

**Fig. 5 | Sex-biased methylation quantitative trait associations in placenta. a** Pie chart displaying the distribution of sex-biased methylation quantitative trait associations based on allelic effect magnitude and direction in males and females (Blue = concordant effect direction, $n = 134$; Green = opposite effect direction, $n = 141$; Orange = single sex effect, $n = 1426$; Gray = Unclassified, $n = 138$). "Unclassified" includes associations that do not fall into the other categories (i.e., sex-biased mQTL associations with two-sided $t$ test FDR-adjusted $P \geq 0.05$ ($n = 291$, df = 289) in both sex strata and/or those with two-sided $t$ test with unequal variance FDR-adjusted $P_{diff} \geq 0.05$ ($n = 291$, df = 289). **b** Regional plot of top male-specific sex-mQTL (rs12242275-cg09082518). The plot to the right indicates the absence of effect in

females ($r_s = 0.02$, $P = 0.91$). *FNDC5* is a precursor of irisin, which protects the placenta from oxidative stress, dysregulation of placental trophoblast differentiation, and insulin resistance[56,57]. Lower irisin levels have been linked to preeclampsia[56,57], a placenta insufficiency-

effect in females. **c** Regional plot of top female-specific sex-mQTL (rs4351362-cp14735364). The plot to the right indicates the absence of effect in males. Data span 200 kb centered at the mQTL SNP. The horizontal axis denotes the genomic position in build hg19, and the vertical axis denotes the association -log10 $P$-value and recombination rate (cM/Mb). The purple circle point represents the mQTL SNP. The purple triangle point represents the mQTL CpG. The color of each data point indicates its linkage disequilibrium value (r²) with the index SNP based on Hap-Map2. LocusZoom (http://locuszoom.org/) was used to generate the plot. Source data for Fig. 5b, c are provided as a Source Data file.

disorder correlated with smaller placental weight[58]. In our data, increased cg25364822 methylation and decreased *FNDC5* expression were correlated with smaller placental weight ($r_s = -0.12$, $P = 0.04$; $r_s = 0.29$, $P = 0.01$, respectively), suggesting a sex-biased epigenetic

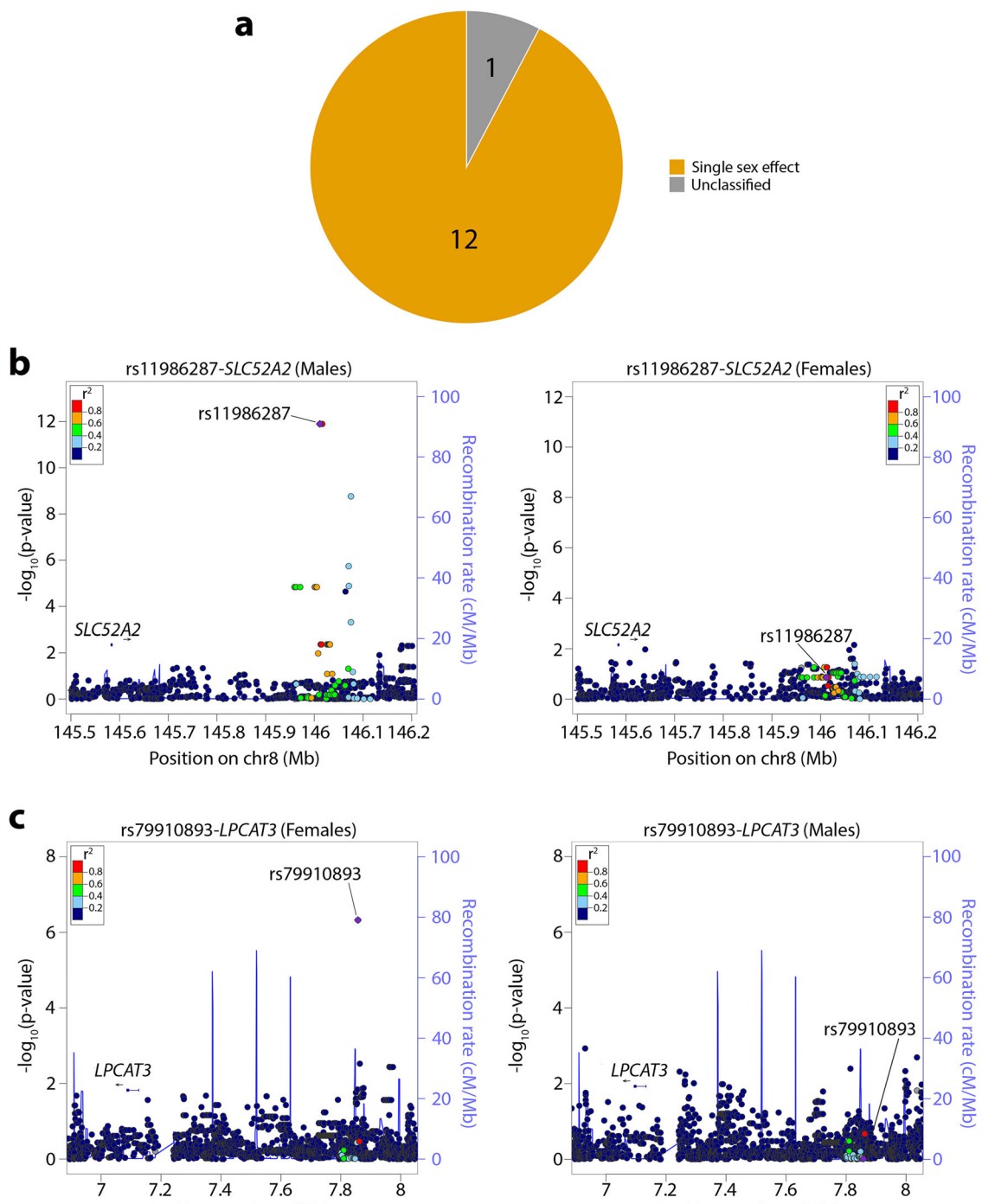

**Fig. 6 | Sex-biased expression quantitative trait associations in placenta. a** Pie chart displaying the distribution of sex-biased expression quantitative trait associations based on allelic effect magnitude and direction in males and females (Orange = single sex effect, $n = 12$; Gray = Unclassified, $n = 1$). "Unclassified" includes associations that do not fall into the other categories (i.e., sex-biased eQTL associations with two-sided t-test FDR-adjusted $P \geq 0.05$ ($n = 71$, df = 69) in both sex strata and/or those with two-sided t-test with unequal variance FDR-adjusted $P_{diff} \geq 0.05$ ($n = 71$, df = 69). **b** Regional plot of top male-specific sex-eQTL (rs11986287-*SLCS2A2*). The plot to the right indicates the absence of effect in females. **c** Regional plot of top female-specific sex-eQTL (rs79910893-*LPCAT3*). The plot to the right indicates the absence of effect in males. Data span 200 kb north of the gene TSS position to 200 kb south of the eQTL SNP. The horizontal axis denotes the genomic position in build hg19, and the vertical axis denotes the association -log10 *P*-value and recombination rate (cM/Mb). The purple circle point represents the eQTL SNP. The color of each data point indicates its linkage disequilibrium value ($r^2$) with the index SNP based on HapMap2. A gene symbol and associated track from the UCSC Genome Browser shows the eQTL target gene's physical location. LocusZoom (http://locuszoom.org/) was used to generate the plot. Source data for Fig. 6b, c are provided as a Source Data file.

process that may regulate gene expression levels that impact placental function. (ii) A co-occurrence of sex-DM and sex-mQTL: male-biased cg27576576 methylation and male-biased association of the G allele of rs6459811 (*PTPRN2*) with higher cg27576576 methylation. The rarity of

overlaps suggests that sex-QTL effects in the placenta are not largely explained by or accompanied with sex differences in methylation or sex differences in gene expression. Similarly, distinct genetic loci have been implicated in sex-biased eQTL and sex-biased gene expression in

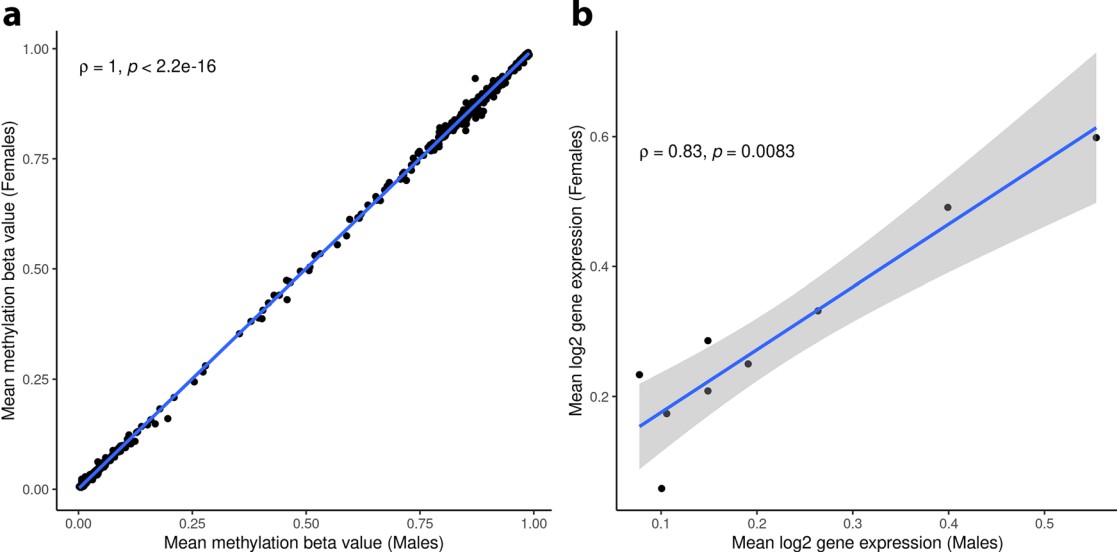

**Fig. 7 | Correlation between males and females of placental methylation and gene expression levels of CpG and gene targets of sex-biased genetic regulation. a** DNA methylation level of sex-mQTL target CpGs (*n* = 521 CpGs). **b** Gene expression levels of sex-eQTL target genes (*n* = 9 gene transcripts). A two-sided

Spearman correlation (ρ) test was implemented and was considered significant if *P* < 0.05. The *p*-value in a is approximately equal to zero. The error band in gray denotes a 95% confidence interval. Source data for Fig. 7a, b are provided as a Source Data file.

other tissues[50,59]. Furthermore, we found significantly high correlations in expression levels of sex-eQTL target genes between males and females ($r_s$ = 0.83, *P* = 0.0083) and in methylation levels of sex-mQTL target CpGs ($r_s$ = 1, *P* < 2.2 × 10$^{-16}$) (Fig. 7a, b), suggesting largely non-overlapping sex-dependent placental methylation and gene expression.

## Placental cell types in sex-biased methylation and gene expression

As the placenta is a cell-heterogenous organ, we assessed whether the sex-QTL associations were due to sex differences in placental cell composition. We found no significant sex difference in placental cell type proportion derived in silico using methylation data[60] or RNA-seq data[61] (Supplementary Data 8). Second, cell type proportions were correlated with the top three methylation principal components (Supplementary Fig. 1) and top three RNA principal components (Supplementary Fig. 2), which were regressed out of the methylation and expression data, respectively, prior to sex-QTL analysis. Third, expression variance components (i.e., PEER factors), which may capture cell type signals were regressed out prior to the sex-eQTL analysis. Therefore, the sex-QTLs identified were unlikely to arise from sex differences in placental cell composition.

## Phenotypic relevance of sex-biased methylation and gene expression

Male-hypermethylated sex-DM CpGs in the placenta were more likely than female-hypermethylated sex-DM CpGs to be positively correlated (*P* < 0.05) with neonatal weight, length, or head circumference (48.4% vs. 0.8%, chi-squared test *P* = 2.61 × 10$^{-321}$), whereas female-hypermethylated sex-DM were more likely than male-hypermethylated sex-DM and to be positively correlated with placental weight or placental weight/birth weight ratio (20.3% *vs.* 3.5%, chi-squared test *P* = 2.11 × 10$^{-207}$) (Supplementary Data 2and Fig. 8). Among male-hypermethylated CpGs, ten were positively correlated with all three neonatal anthropometry measures and negatively correlated with placenta-birthweight ratio (CpGs map to *EGLN1*, *FNBP1L*, *TSPAN14*, *TEAD1*, *MIR130A*, *TMEM223*, *DOCK9*, *SLC1A5*, *DGCR6L*, and *ST6GALC6* genes). *EGLN1* encodes a protein in the hypoxia-inducible factor (HIF) pathway and is important for placental development

through its role as an oxygen sensor and regulator of *HIF1A* expression in trophoblast cells[62,63]. Among female-hypermethylated CpGs, three were positively correlated with placenta-birth weight ratio and negatively correlated with at least two neonatal anthropometry measures (CpGs map to *SHANK3*, *POP4*, and *GRHL3* genes). *SHANK3* encodes a synaptic protein that is crucial for the development of brain circuits[64]. The *SHANK3* gene locus has been implicated in cognitive function measurement and schizophrenia[65,66] and in autism spectrum disorders[64], which is more commonly diagnosed in males than females[67]. Placental hypomethylation at a nearby locus (22q13.33) has been linked with autism[68], and hypomethylation of our female-hypermethylated CpG in adult blood DNA has been associated with brain gray matter volume[69].

We assessed whether the sex-biased loci have previously been implicated in fetoplacental growth. A male-hypermethylated sex-DM at cg10806146 (5′UTR of *SLC20A2*) has previously been associated with higher birthweight[70]. *Slc20a2* is required for normal placental phosphate transport function in mice, and its deficiency has been linked with defective vascularization, increased calcification of the placenta, and fetal growth restriction[71]. In addition, three genes near sex-DM CpGs (*HSPA4*, *SLC45A4*, and *SLC6A2*) and two genes near sex-mQTL CpGs (*SLC45A4* and *ARHGAP26*) map fetal genetic variants associated with placental weight in previous GWAS[72].

Further, we looked up the sex-biased loci in the EWAS (epigenome-wide association study) Atlas, EWAS Catalog, and GWAS Catalog. Several sex-DM and sex-mQTL CpGs overlapped with CpGs in blood previously reported to be associated with pregnancy complications (e.g., preterm birth, preeclampsia, gestational diabetes mellitus), fetal and child growth, adult complex diseases (e.g., aging), and prenatal environmental exposures. Sex-mQTL genetic variants, genes near sex-DM CpGs, and sex-DE genes have links with GWAS traits in adults (Supplementary Data 9).

## Colocalization of sex-QTLs with birthweight and adult diseases

To explore the usefulness of sex-QTLs in understanding the molecular basis of complex trait GWAS discoveries, we performed colocalization analysis by integrating sex-stratified QTL in the placenta with GWAS. Among sex-mQTLs, we identified 18 CpGs colocalized with 19 distinct GWAS loci for birthweight (eCAVIAR posterior probability of sharing

### a  Positive Correlation, Male-hypermethylated

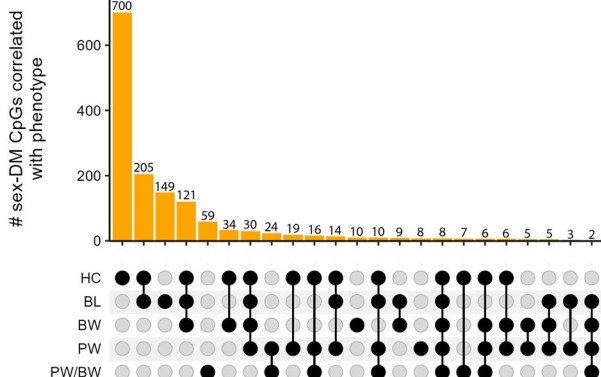

### b  Positive Correlation, Female-hypermethylated

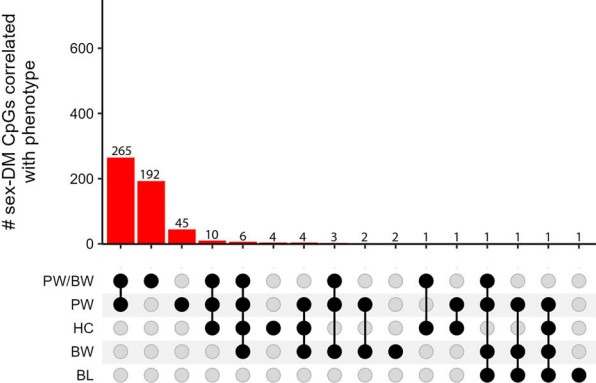

### c  Negative Correlation, Male-hypermethylated

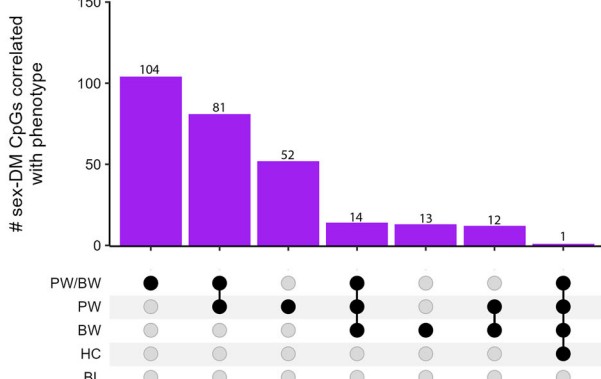

### d  Negative Correlation, Female-hypermethylated

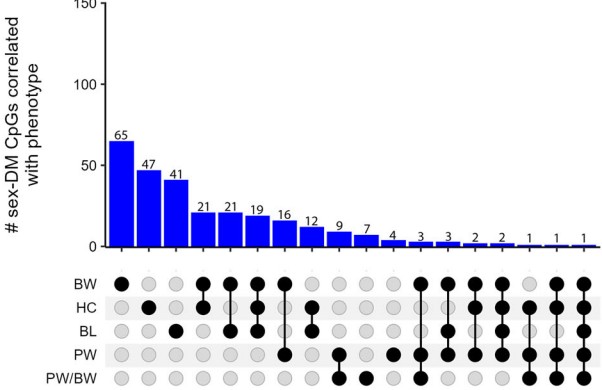

**Fig. 8 | Upset plot of male- and female-hypermethylated DNA methylation sites showing overlapping correlation with neonatal anthropometry and placental size measures.** The figures display the number of male-hypermethylated (**a**) and female-hypermethylated (**b**) CpG sites that are positively correlated with neonatal and placental measures, and the number of male-hypermethylated (**c**) and female-hypermethylated (**d**) CpG sites that are negatively correlated with neonatal and placental measures. The measures are head circumference (HC), birth length (BL), birth weight (BW), placental weight (PW), and placental-birth weight ratio (PW/BW). Vertical lines connecting black-shaded circles denote measures that exhibited a shared correlation with the number of CpGs in bars.

the same causal variant ≥ 0.01), revealing the importance of sex-differentiated epigenetic mechanisms in the placenta to interpret genetic associations with fetal growth, a sex-differentiated trait tightly regulated by the placenta[2]. The strongest colocalizations were between cg10611863 (*ENTPD4*) and birthweight for female-stratified mQTL, and between cg02370022 (*CCND1*) and birthweight for male-stratified mQTL. Among sex-eQTLs, we identified 4 colocalized gene-adult trait pairs, of which female-stratified eQTLs at *ATP5MG* and *FAM83A* colocalized with asthma/hay fever/eczema and breast cancer and male-stratified eQTLs at *FAM103A2P* and *SLC52A2* colocalized with breast cancer and educational attainment, respectively (Supplementary Data 10).

### Enrichment for imprinted genes

Imprinting, an epigenetic regulation of gene expression in a parent-of-origin-dependent manner, has well-known roles in regulating placental development and function[22,73]. Sex bias has been reported in the regulation of imprinted genes in rodent placenta[74,75], so we tested whether there is enrichment of imprinting regions among the sex-DM. We found significant enrichment of imprinted genes in sex-DM genes (61 imprinted genes out of 2225 unique sex-DM genes compared to 228 known imprinted genes in the human genome; hypergeometric test $P = 2.599 \times 10^{-9}$). The largest regional cluster of sex-DM CpGs was a 63 kb region in the maternally expressed *TP73* gene and the homeobox family genes (*HOXA3*, *HOXA4*, *HOXA5*, *HOXB2*, *HOXB3*, and *HOXC9*),

known for their crucial roles in regulating placental and fetal development[76]. Thirteen of the 61 imprinted genes are known to be imprinted specifically in the placenta only[77] (Table 1 and Supplementary Data 11).

The identified imprinting overlaps included a cluster of 13 female-hypermethylated sex-DM CpGs within the well-known *KCNQ1OT1/CDKN1C* imprinting domain in chromosome 11p15.5 genomic region. In the human placenta, the *KCNQ1OT1/CDKN1C* imprinting domain harbors genes such as *CDKN1C*, *PHLDA2*, and *SLC22A18* that are maternally expressed in the first trimester and term placenta, a long non-coding RNA (*KCNQ1OT1/kvDMR1*) with paternal expression in the first trimester and biallelic expression in term placenta, and other genes such as *KCNQ1* which exhibits maternal expression in first-trimester placenta and biallelic expression in term placenta[78]. The imprinting domain is regulated by a maternally methylated DMR imprinting control region (ICR) located in intron 10 of the *KCNQ1* gene[78]. A genetic variant in intron 10 of *KCNQ1* has shown a maternal parent-of-origin effect on placental weight[72]. Out of the 13 sex-DM CpGs, 8 were located within *KCNQ1* at 13.7–165.1 kb distance from the ICR and overlap with histone 3 lysine 27 trimethylation (H3K27me3), a posttranslational epigenetic modification associated with transcriptional repression. Given these annotations, we assessed whether methylation at the 8 CpGs has a potential sex-dependent relationship with the placental expression of genes in the chr11p15.5 imprinting domain. Higher CpG methylation showed correlation with increased *KCNQ1OT1* expression in females

**Table 1 | Placenta-specific imprinted genes that overlap with genes near sex-differentially methylated sites in placenta**

| Sex-DM CpG site | Sex with hypermethylation | Imprinted gene (position) | Expressed allele | UCSC RefGene | Evidence for imprinting |
|---|---|---|---|---|---|
| cg22354234 | Female | THUMPD2 (2p22.1) | Maternal | TSS200 | PMID: 27843122 |
| cg18886444 | Female | USP4 (3p21.31) | Paternal | TSS1500 | PMID: 27835649 |
| cg03902093 | Male | RAB7A (3q21.3) | Unknown | TSS1500 | PMID: 32324732 |
| cg10189695 | Male | GPR78 (4p16.1) | Paternal | 1st Exon | PMID: 27835649 |
| cg11229771 | Male | PDE6B (4p16.3) | Paternal | Body | PMID: 27835649 |
| cg06621682 | Male | GRID2 (4q22.1-22.2) | Paternal | Body | PMID: 27835649 |
| cg21186296 | Male | PTK2B (8p21.2) | Unknown | 5'UTR | PMID: 32324732 |
| cg10594837 | Female | DENND3 (8q24.3) | Paternal | Body | PMID: 27835649 |
| cg14984684 | Male | RASGRF1 (15q24.2) | Paternal | 1st Exon | geneimprint.com |
| cg23846509 | Female | LRRK1 (15q26.3) | Unknown | Body | PMID: 32324732 |
| cg16950726 | Female | NLGN2 (17p13.1) | Paternal | Body | PMID: 27843122 |
| cg13105709 | Female | SEPTIN4 (17q22) | Paternal | 3'UTR | PMID: 32324732 |
| cg20042338 | Male | ANO8 (19p13.11) | Maternal | Body | PMID: 32324732 |

*Sex-DM CpG* sex-differentially methylated cytosine-phosphate-guanine sites (CpG), *UCSC RefGene* University of California Santa Cruz reference sequence annotation of the CpG sites in relation to the corresponding imprinted gene, *TSS* transcription start site, *UTR* untranslated region.

and trended with decreased *KCNQ1OT1* expression in males (cg03030994-*KCNQ1OT1*: $r_m = -0.26$, $P_m = 0.12$; $r_f = 0.36$, $P_f = 0.04$), increased *CDKN1C* expression in females (cg00446023-*CDKN1C*: $r_m = -0.09$, $P_m = 0.58$, $r_f = 0.35$, $P_f = 0.046$), and decreased *SLC22A18* and *KCNQ1* expression in males (cg05457684-*SLC22A18*: $r_m = -0.44$, $P_m = 0.006$, $r_f = 0.04$, $P_f = 0.83$; cg05457684-*KCNQ1*: $r_m = -0.46$, $P_m = 0.004$; $r_f = 0.05$, $P_f = 0.79$; cg06960356-*KCNQ1*: $r_m = -0.42$, $P_m = 0.009$; $r_f = 0.14$, $P_f = 0.45$). The *KCNQ1OT1/CDKN1C* domain has been linked to diseases and trait differences. For example, imprinting dysregulation of genes in the *KCNQ1OT1/CDKN1C* domain has been linked to Beckwith-Wiedemann syndrome, a disorder of growth regulation characterized by somatic overgrowth and tumor predisposition[79]. Moreover, increased placental expression of the maternally expressed *CDKN1C*, *PHLDA2*, and *SLC22A18* genes has been linked with fetal growth restriction and smaller neonatal size[80–84]. We found that three female hypermethylated sex-DM CpGs showed correlation, albeit weakly, with lower birthweight (cg25548316: $r_s = -0.13$, $P = 0.03$), higher placental weight (cg13536051: $r_s = 0.11$, $P = 0.048$; cg15782852: $r_s = 0.18$, $P = 0.002$), and higher placenta-birthweight ratio (cg13536051: $r_s = 0.2$, $P = 0.001$; cg15782852: $r_s = 0.14$, $P = 0.02$).

### Enrichment of pathways and hallmark gene sets

Most canonical pathways and hallmark gene sets that were enriched for genes near male-hypermethylated sex-DM CpGs differed from those enriched for female-hypermethylated sex-DM CpGs (Fig. 9, Supplementary Fig. 3 and Supplementary Data 12). The top canonical pathways enriched only in male-biased methylation included extracellular matrix organization, hemostasis, and immune response (e.g., chemokine and Family B G-protein-coupled receptors); pathways enriched only in female-biased methylation included transcription regulation, transportation of small molecules in cells, and regulation of genes involved in metabolism by the tumor suppressor protein TP53. The top hallmark gene sets enriched only in male-biased methylation included coagulation, epithelial-mesenchymal transition, upregulation of KRAS signaling, myogenesis, and allograft rejection; hallmark gene sets enriched only in female-biased methylation included mitotic spindle, early estrogen response, apoptosis, and glycolysis. For the minority of pathways and hallmark gene sets that overlapped between male- and female-hypermethylated sex-DM CpGs, few or no genes were common between males and females. Such gene set convergence through different genes suggests that male and female placentas may coordinate different CpGs/genes for an essentially identical biological process or potential compensatory mechanisms.

### Enrichment for transcription factor binding sites

Transcription factor (TF) activity can contribute to sex-biased gene regulation[55]. We found that several transcription factor binding sites (TFBS) are enriched for the sex-biased methylation loci. The identified TFBS are associated with TFs that regulate target genes through diverse mechanisms. Specifically, the sex-mQTL genetic variants were significantly enriched for three TFBS associated with TFs that regulate target genes via functions such as nuclear hormone receptors (e.g., *NR2C2*), sequence-specific DNA binding (*SP2*), and epigenetic remodeling (*REST*) (FDR-adjusted $P < 0.05$). *NR2C2* (nuclear receptor subfamily 2 group C member 2) encodes the testicular receptor 4 protein TR4. In vivo mice studies have shown that *Nr2c2* plays key roles in fetal growth, early postnatal survival, fertility, and sensitivity to environmental stimuli[85,86]. Male *Nr2c2*-null mice exhibit delayed spermatogenesis and reduced fertility[86], and females exhibit maternal behavioral abnormalities[85]. Sex-DM and sex-mQTL CpG genomic positions were significantly enriched for TFBS associated with TFs in the zinc finger family, cell cycle regulation (*E2F*), sequence-specific DNA binding (*THAP*), site-specific DNA binding and embryogenesis (*AP-2*) (Supplementary Fig. 4).

## Discussion

This study offered several unique insights about the landscape of sex differences in the level and genetic regulation of methylation and gene expression in the human placenta. Key findings include the following: (i) Placental autosomal methylation and its genetic regulation are extensively impacted by fetal sex, whereas sex differences in both the level and the genetic regulation of placental autosomal gene expression are limited. (ii) The vast majority of placental methylation sites and gene transcripts impacted by sex-biased genetic effects did not exhibit methylation and gene expression differences by sex. Moreover, methylation sites that were sex-biased were not enriched for predicted TF binding, did not largely correlate with annotated gene expression, and were enriched for molecular pathways largely distinct from previously described pathways related to sex-biased gene expression in placenta[14,17]. These observations suggest that distinct mechanisms may underlie sex-dependent placental methylation, gene expression, and their genetic regulation. (iii) The sex-biased methylation sites overlapped with genetic loci implicated in several human phenotypes. Notably, several sex-differentiated genetic effects on methylation and gene expression in the placenta colocalized with birthweight, and adult traits such as breast cancer and allergic diseases, contributing to the mechanistic interpretation of GWAS signals. These findings

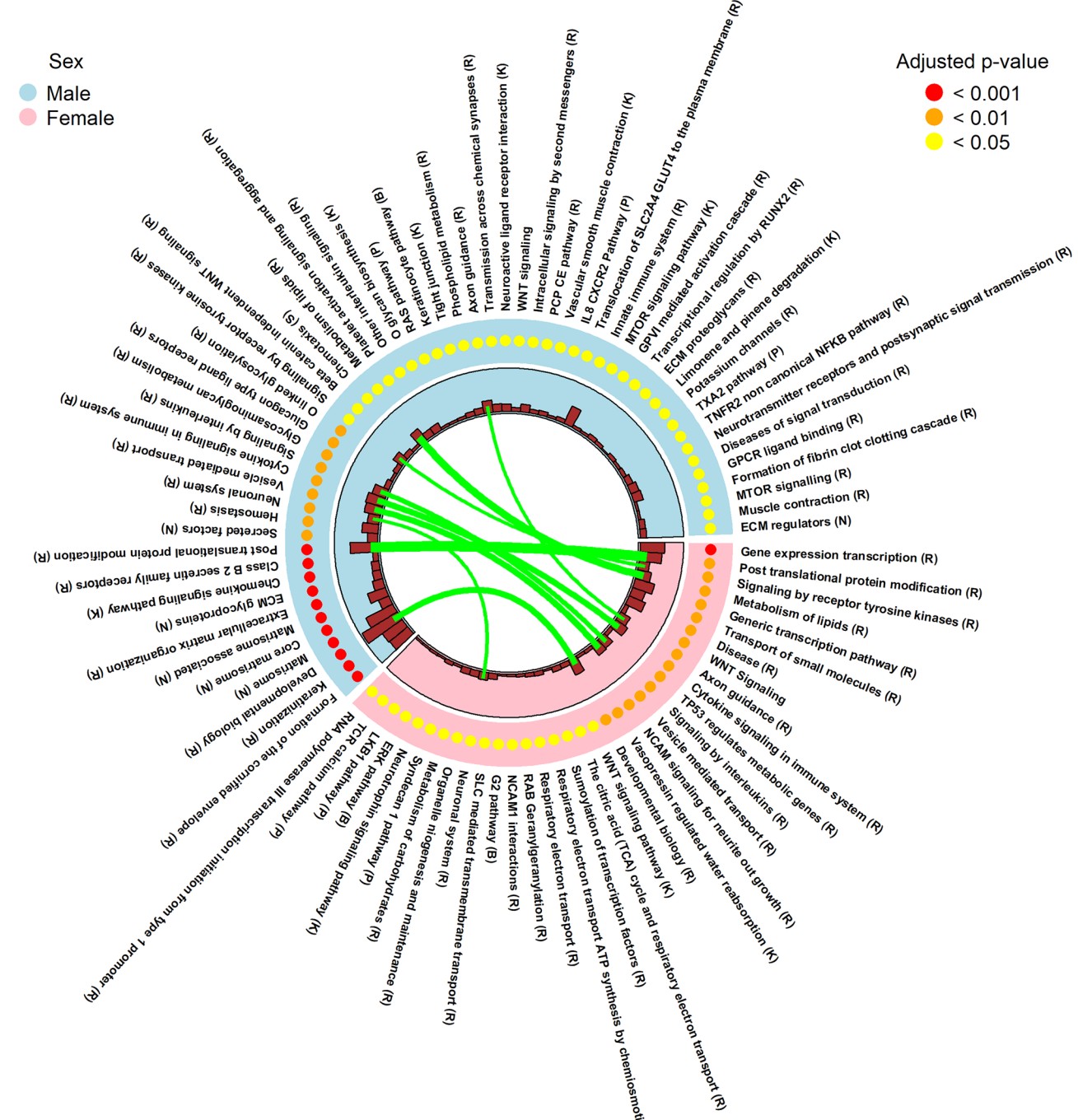

**Fig. 9 | Circos plots of significantly enriched canonical pathways for genes near sex-differentially methylated sites.** The outer track represents the enriched canonical pathways obtained using the FUMA tool, ordered from the most to the least significant in the clockwise direction. The middle track represents the hypergeometric test for enrichment FDR-adjusted *P*-value. The orange bars in the inner-most track represent the number of genes near the sex-differentially methylated sites that overlapped with genes in the database of each pathway. The green lines connect enriched pathways shared between males and females, with width proportional to the number of shared genes. Database abbreviations: B: BIOCARTA, K: KEGG, N: NABA, P: PID, R: REACTOME, S: SIG. Source data are provided as a Source Data file.

suggest the potential role of placenta-mediated sex differences in developmental and later-life physiological traits and diseases.

Many genetic variants showing sex-biased association with placental methylation and gene expression had a single-sex effect, unveiling biological insights that would be masked or attenuated in standard combined-sex QTL studies. One such example is an mQTL near *STMN3* (Stathmin-3) (rs1151625-cg14862171), identified previously by a combined-sex analysis of placenta samples[87], and was found in the present study to be a sex-mQTL locus conforming to a female-specific

effect. The genetic variant's effect on methylation was 45-fold stronger in females than male placenta and 1.29-fold stronger in female than combined-sex sample placenta. These results clarify that the previously reported association based on combined-sex analysis[87] was most likely driven by the effect of the genetic variant on methylation in the female placenta only. *STMN3* encodes a microtubule destabilizing protein that regulates placentation in early pregnancy in vitro[88]. The largely distinct sex-DM and sex-DE associations detected in the present study align with a recent report[20]. Our study implemented more omics-

integrated analyses, including directly testing association of sex-DM CpGs with nearby gene expression, identifying sex-eQTL associations, and implementing analyses models that included interaction terms between sex and genetic variants. These approaches yielded unique insights that sex-DM CpGs were more likely to be associated with nearby gene expression alteration and enabled the detection of several previously undetected sex-mQTLs and sex-eQTLs with single-sex effects.

Male fetus-bearing pregnancies are at increased risk for major pregnancy complications such as pre-eclampsia, placental abruption, fetal growth restriction, and post-partum hemorrhage, all of which have molecular origins tied to placentation and hemostasis[4]. In our study, male-biased placental methylation was markedly enriched for the hemostasis/coagulation hallmark gene set that has not previously been characterized as sex-differentiated. The placenta expresses coagulation components involved in hemostasis and placental vascular development[89]. The male-enriched coagulation hallmark gene set included tissue factor protein inhibitor-2 (TFPI2; aka placental protein 5 (PP5)), an extracellular matrix-associated protein known to be abundantly produced by the placenta and is actively involved in coagulation and hemostasis processes[90]. Transition to a state of hyper-coagulability is maximal around the term to prevent maternal and child health complications due to excessive bleeding[91]. Our finding suggests that epigenetically induced suppression of coagulation pathways can be one potential explanation for male preponderance in pregnancy pathologies and contributes to emerging knowledge on coagulation-inflammation/immune response crosstalk during pregnancy.

We found a previously unrecognized enrichment of imprinting genes in sex-differentiated placental methylation, including a cluster of female-biased methylation within the *KCNQ1OT1/CDKN1C* imprinting domain implicated in Beckwith-Wiedemann syndrome[78,79]. In our data, higher methylation of the female-hypermethylated CpGs was associated with female-specific upregulation or male-specific down-regulation of genes in this imprinting domain. These findings suggest a potential crosstalk between sex and imprinting regulation in the placenta. The unique epigenetic architecture of imprinting in the human placenta may be permissive to epigenetic alterations. For example, in the *KCNQ1OT1/CDKN1C* cluster, loss of imprinting is variable, and gene repression by the *KCNQ1OT1* lncRNA is incomplete[92]. Human placental DMRs are uniquely highly polymorphic[73,78,93]. Placental DMRs are associated with variable histone marks[94] which are mechanistically linked to DNA methylation that influences imprinting stability[95] and potentially with underlying genetic variation[93]. Collectively, these studies and our findings suggest that the imprinting domains may be vulnerable to epigenetic alterations from other factors that influence allelic bias in a sex-dependent manner, but future work should investigate this and its relevance to sex differences in pregnancy physiology and clinical complications.

Our study has limitations. First, placental gene expression and methylation are dynamic over gestation[16,17]. However, the current study is unable to distinguish sex differences that emerge in early gestation from those that emerge in later gestation because the placenta samples were obtained at or near-term gestation. Second, genetic regulation of methylation and gene expression in the placenta may vary by cell type. We found that in silico-derived placental cell type composition did not vary by sex and did not explain the observed sex-QTLs. Similar to ours, the GTEx study did not find sex differences in cell composition for 41 out of 44 tissues tested[50]. These findings should be seen in light of the caveat that existing *in-silico* methods for estimating the cell type composition of the placenta do not fully characterize the spatial heterogeneity of the placenta. Better insights may be gained in the future through single-cell analysis of the placenta from different developmental stages. Third, differences related to sex chromosomes will require future methodological advances that can

facilitate analyses that integrate the autosomes with the haploid Y chromosome and dosage compensation of the X chromosome[1]. Fourth, the detection of sex-eQTLs may have been limited by the sample size of our placental RNA-seq dataset. The inclusion of covariates to account for genetic population structure and cell heterogeneity may have further impacted the study's power in detecting weaker signals. Similar to our findings, the limited sex difference in gene expression regulation has been observed in 44 non-placental tissues from GTEx[50] and in adult blood[96,97]. Moreover, one study demonstrated that samples of millions of individuals would be required to detect sex-biased eQTLs in the blood that appreciably mediate sex-biased genetic associations with complex traits[96]. Larger datasets and accounting for factors that influence QTLs can improve the transferability of sex-QTLs found to be limited in our study as well as others[50,98].

In conclusion, this multi-omics study in the placenta identified sex-specific regulatory processes and molecular pathways and potential crosstalk between placental sex and imprinting. Sex-biased genetic regulation of placental methylation and gene expression colocalized with GWAS loci for neonatal traits and adult diseases, revealing placental embedding of sex differences in human health and disease across the life span.

## Methods

The NICHD Fetal Growth study protocol was approved by the institutional review boards of NICHD and each of the participating clinic sites, namely, Columbia University, New York; New York Hospital, Queens, New York; Christiana Care Health System, Delaware; Saint Peter's University Hospital, New Jersey; Medical University of South Carolina, South Carolina; University of Alabama, Alabama; Northwestern University, Illinois; Long Beach Memorial Medical Center, California; University of California, Irvine, California; Fountain Valley Hospital, California; Women and Infants Hospital of Rhode Island, Rhode Island; and Tufts University, Massachusetts. The RICHS study was approved by institutional review boards of Emory University and Women and Infants Hospital of Rhode Island.

### Dataset

Datasets from placenta samples obtained at delivery as part of the *Eunice Kennedy Shriver* National Institute of Child Health and Human Development (NICHD) Fetal Growth Studies – Singletons were included in the discovery analysis. The NICHD Fetal Growth Studies – Singletons recruited 2802 pregnant women between July 2009 and January 2013 at 12 clinical sites in the United States from four race/ethnic groups (i.e., non-Hispanic White, non-Hispanic Black, Hispanic, and Asian or Pacific Islander) between 8–13 gestational weeks and followed through delivery. To be enrolled, women had to have no past adverse pregnancy outcomes and no major pre-existing medical conditions, including autoimmune diseases, chronic hypertension, diabetes, chronic renal disease, cancer, HIV/AIDS, or psychiatric disorders. Gestational age was determined using the date of the last menstrual period and confirmed by ultrasound between 8 weeks to 13 weeks and 6 days of gestation. After the first ultrasound, pregnant women underwent up to five standardized ultrasounds with measurement of fetal biometry at a priori-defined gestational ages[25]. Birth weight was measured in grams (g) using an electronic infant scale or beam balance scale, and birth length and head circumference were measured in centimeters (cm). Placenta weight was measured in grams (g). Written informed consent was obtained from all study participants.

### Genotyping, methylation and RNA-seq procedures and quality control

Placental biopsies measuring 0.5 cm × 0.5 cm × 0.5 cm were taken from the fetal side (*n* = 312) within one hour of delivery, and samples were

placed in RNAlater and frozen in − 80 °C for molecular analysis[87]. DNA extracted from the placental biopsies was genotyped using HumanOmni2.5 Beadchip (Illumina Inc., San Diego, CA). For the genotype dataset, detailed quality control procedures have been described previously[21,87,99,100]. Briefly, single nucleotide polymorphisms (SNPs) with >5% missing values, minor allele frequency <0.5%, and not in Hardy-Weinberg equilibrium ($P < 10^{-4}$) we removed. SNP genotypes were imputed using the Michigan Imputation Server (https://imputationserver.sph.umich.edu/) using the 1000 Genomes Phase 3 genotype reference (https://www.internationalgenome.org/category/reference/), and filters were applied to remove insertion-deletions, SNPs with minor allele frequency < 0.5% and SNPs with imputation dosage $r^2 < 0.3$. Infant sex (male or female) was obtained from medical charts and was compared with sex predicted based on X chromosome heterozygosity (inbreeding coefficient estimate $F > 0.8$ male, $F < 0.2$ female)[21,87,99,100].

DNA methylation was profiled on the 312 samples using Illumina's Infinium Human Methylation450 Beadchip (Illumina Inc., San Diego, CA). Probes which were cross-reactive, non-autosomal, had mean detection $P \geq 0.05$, or had CpGs located within 20 base pairs of known SNPs were removed. Samples that had discrepancies between sex obtained from medical charts and predicted from genotypes, were outliers from the distribution of the samples' genetic clusters or had a mismatching sample identifier were removed, as previously described[21,87,99,100].

RNA from a subset of the placental samples ($n = 80$) was extracted using TRIZOL reagent (Invitrogen, MA), and RNA sequencing was performed using the Illumina HiSeq2000 system with 100 bp paired-end reads. The reads were mapped to the human reference genome (NCBI/build 37.2) using Tophat version 2.0.4. The raw placental RNA-seq dataset had a total of 33,690 RNA transcripts. Read count data were normalized across libraries using the trimmed mean of M values (TMM) normalization technique implemented in the R/Bioconductor package "edgeR"[101]. Transcripts in autosomal chromosomes without 6 or more reads in at least 3 samples or without depth-normalized counts per million (CPM) of 0.1 or higher in at least 3 samples were removed, as previously described[22,87].

The following datasets that passed previously performed quality control filters mentioned above were included in the present analyses: 301 samples (152 males and 149 females) with placental methylation data at 408,680 CpGs were included in sex-DM analysis, of which 291 samples (147 males and 144 females) with fetal genotype data at 5,359,103 genetic variants as well as placental methylation data at 408,680 CpGs were included in sex-mQTL analysis. A total of 80 sub-samples (43 males and 37 females) with placental RNA-seq data at 21,550 transcripts were included in sex-DE analysis, of which 71 samples (38 males and 33 females) with fetal genotype data at 5,337,343 genetic variants as well as RNA-seq at 21,550 transcripts were included in sex-eQTL analysis.

## Sex-biased methylation differentiation (sex-DM) analyses

A total of 301 samples (152 males and 149 females) with placental methylation data at 408,680 CpGs were included in the sex-DM analysis. We estimated placental cell type proportion in silico using DNA methylation beta values with the R/Bioconductor package "planet" (https://www.bioconductor.org/packages/release/bioc/html/planet.html)[60]. Genotype-based principal components (PCs) representing population structure were estimated using fetal genome-wide SNP genotype data. Methylation PCs were estimated using the R package "prcomp"[102]. To identify sex-differentiated methylation (sex-DM), epigenome-wide analyses were performed with fetal sex as the predictor and placenta DNA methylation at each CpG site as the outcome using the R/Bioconductor package "limma"[103]. The analysis included linear regression models that were adjusted for self-reported maternal race/ethnicity (non-Hispanic White, non-Hispanic Black, Hispanic,

Asian), gestational age at delivery, methylation sample plate ($n = 5$), the first three methylation PCs, the first 10 genotype PCs to account for population structure, and predicted cell type proportions for six placental cell types[60]. To account for the inflation of statistical tests, we implemented a Bayesian method to obtain BACON-corrected inflation estimates and BACON-corrected $P$-values using the R/Bioconductor package "BACON"[104]. Quantile-quantile (QQ) plots of $P$-values and the corresponding inflation estimate after BACON-correction ($\lambda = 1.271$) are reported in Fig. 2a. BACON-corrected $P$-values were then controlled for false discovery rate (FDR), giving BACON-corrected FDR-adjusted $P$-values.

## Correlation between sex-DM CpG methylation and gene expression

To evaluate relations between sex-DM CpGs and gene expression, we tested whether methylation at each sex-DM CpG is correlated with the placental expression of its closest gene. This test was performed between 3636 sex-DM CpGs located in or near a gene (i.e., in gene body, 3'untranslated region (UTR), 5'UTR, exon, transcription start site (TSS) based on the University of California Santa Cruz (UCSC) genome reference annotation; https://genome.ucsc.edu/cgi-bin/hgGateway[105]) and placenta nearby gene expression levels in our dataset. In further exploration, we assessed the correlation between methylation at each sex-DM CpG and placental expression of all genes within 200 kb distance from the CpG site.

## Sex-biased cis-mQTL (sex-mQTL) analyses

A total of 291 samples (147 males and 144 females) with both placental methylation data at 408,680 CpGs and fetal genotype data at 5,359,103 genetic variants were included in the sex-mQTL analysis. Sex-mQTL analysis was performed to identify sex-mQTLs, i.e., SNPs whose effect on placental methylation differs in magnitude between male and female pregnancies. The sex-mQTL analysis was performed in two stages to accommodate the computational burden of the model for analysis in MatrixEQTL[106]. First, residuals of each methylation site were obtained by running a linear regression model with each methylation site as an independent outcome and race/ethnicity, gestational age at delivery, methylation plate, 3 methylation PCs, and 10 genotype PCs as covariates. Next, each methylation residual was fit as the outcome in a linear regression model with SNPs within 1 Mb distance and an interaction term between SNP and sex using MatrixEQTL[106]. The QQ plot of $P$-values for the SNP*sex interaction term of the mQTL mapping showed the absence of inflation ($\lambda = 1.001$) (Fig. 2b). Statistical significance for the interaction term was defined based on FDR-adjusted $P < 0.05$.

Next, sex-stratified cis-mQTL analyses were performed on the FDR-significant mQTL-CpG pairs in a linear regression model with methylation as outcome, SNP within 1 Mb distance as predictor, and race/ethnicity, gestational age at delivery, methylation plate, 10 genotype PCs, and 3 methylation PCs as covariates. For pairs found to have FDR-significant mQTL association in either or both males and females (FDR-adjusted $P < 0.05$, accounting for the total number of mQTL-CpG pairs tested in the two strata), we performed a t-test of the difference between the standardized male-specific and female-specific effect estimates, assuming unequal variance ($P_{diff}$). Lastly, we classified the sex-mQTLs with FDR-adjusted $P_{diff} < 0.05$ into one of the following three categories: (i) concordant effect: association found to be FDR-significant in one sex and nominally significant ($P < 0.05$) or FDR-significant in the other sex with a consistent effect direction but different magnitude; FDR-adjusted $P_{diff} < 0.05$. (ii) opposite effect: association found to be FDR-significant in one sex and nominally significant ($P < 0.05$) or FDR-significant in the other sex a with an opposite effect direction; FDR-adjusted $P_{diff} < 0.05$. (iii) Single-sex effect: association found to be FDR-significant in one sex but not nominally significant ($P < 0.05$) in the other sex; FDR-adjusted $P_{diff} < 0.05$.

### Sex-biased gene expression differentiation (sex-DE) analyses

A total of 80 samples (43 males and 37 females) with placental RNA-seq data at 21,550 transcripts were included in the sex-DE analysis. Analysis was performed using edgeR, adjusted for race/ethnicity, gestational age at delivery, and 10 genotype PCs. Quantile-quantile (QQ) plots of $P$-values and the corresponding inflation estimate after BACON correction ($\lambda = 1.326$) are reported in Fig. 2c.

### Sex-biased cis-eQTL (sex-eQTL) analyses

A total of 71 individuals (38 males and 33 females) with both fetal genotype data at 5,337,343 genetic variants and RNA-seq at 21,550 transcripts were included in sex-eQTL analysis. Sex-eQTL analysis was performed in two stages to accommodate the computational burden of the model for MatrixEQTL[106]. First, residuals of each transcript were obtained by running a linear regression model with each transcript as the independent outcome and race/ethnicity, gestational age at delivery, 10 PEER factors (i.e., expression variance components)[107], 3 RNA PCs, and 10 genotype PCs as covariates. Next, each transcript's residual was fit as an outcome in a linear regression model with SNPs within 1 Mb distance, and an interaction term between SNP and sex using MatrixEQTL[106]. The QQ plot of $P$-values for the SNP*sex interaction term of the eQTL mapping showed the absence of inflation ($\lambda = 0.988$) (Fig. 2d). Statistical significance for the interaction term was defined based on FDR-adjusted $P < 0.05$.

Next, sex-stratified cis-eQTL analyses were performed in a linear regression model with gene expression as an outcome, SNP within 1 Mb distance as the predictor, and race/ethnicity, gestational age at delivery, 3 PEER factors, 4 genotype PCs, and 3 RNA PCs as covariates. The same workflow described for sex-mQTL was implemented to classify the sex-eQTLs with FDR-adjusted $P_{diff} < 0.05$ into concordant effect, opposite effect, or single-sex effect.

### Evaluation of associations in an independent dataset

The sex-DE, sex-mQTL, and sex-eQTL associations discovered were assessed in a dataset from the RICHS cohort ($n = 148$; 74 males, 74 females). Mother and infant pairs were recruited following delivery at the Women and Infants Hospital of Rhode Island from 2009 to 2013. The study included infants born small for gestational age, large for gestational age, and controls born appropriate for gestational age-matched on sex, gestational age, and maternal age[26]. Placental RNA-seq data from a subset of samples ($n = 200$) were obtained using the Illumina Hi-Seq 2500 platform; placental DNA methylation data ($n = 220$) were obtained using the Infinium MethylationEPIC array (Illumina); and genotype data for 159 infants were obtained using the Illumina MegaEX array and imputed using the Haplotype Reference Consortium reference panel. Both sex-mQTL and sex-eQTL analyses were performed on 148 samples (74 male, 74 female) with genotype, DNA methylation data, and RNA-seq data. Of the 1839 SNP-CpG pairs in the sex-mQTL associations discovered in the NICHD Fetal Growth Studies cohort, 778 SNP-CpG pairs were available and tested in RICHS. Of the 14 SNP-gene pairs in the sex-eQTL associations discovered, only 1 pair (rs10892219-ATP5MG) was present and tested in RICHS. Of the 14 sex-DE associations discovered, 12 transcripts with RNA-seq data in RICHS were tested. Sex-QTL analyses were performed using linear regression models as in the discovery analysis with adjustment for self-reported ethnicity, methylation or RNA-seq batch, top 10 genotype PCs, and top 3 expression PCs (for sex-eQTL) or methylation PCs (for sex-mQTL). Sex-DE analyses were performed using linear regression models as in the discovery analysis with adjustment for race/ethnicity, gestational age at delivery, and 10 genotype PCs.

### Placenta cell type proportion

For each of our samples, we derived proportions for 6 placental cell types (trophoblast, stromal, Hofbauer, endothelial, nucleated red blood cells (nRBC), syncytiotrophoblast) using methylation data and a

reference panel of placental cell counts as implemented in the R package "planet" (https://www.bioconductor.org/packages/release/bioc/html/planet.html)[60]. We evaluated whether cell type proportions were different between males and females using a Wilcoxon rank-sum test ($P < 0.05/6$). We further assessed the correlation between cell proportion and methylation PCs 1-3, which were regressed out of the methylation data during sex-mQTL analysis.

For RNA-seq, we also derived proportions for 27 placental cell types with CIBERSORTx[61] based on placental single-cell sequencing reference composed of 19 fetal and 8 maternal cell types[108] and tested whether cell type proportions were different between males and females using a Wilcoxon rank-sum test ($P < 0.05/27$). We further assessed the correlation between cell proportion and RNA PCs 1-3, which were regressed out of the transcript data during sex-eQTL analysis.

### Colocalization analysis between sex-QTL and GWAS loci

To assess whether a shared causal genetic variant underlies both a complex phenotype and sex-biased gene expression/methylation in the placenta, we performed GWAS-QTL colocalization analysis using the ezQTL Web platform (https://analysistools.cancer.gov/ezqtl/#/home)[109]. ezQTL performs colocalization analysis using two methods (eCAVIAR and HyPrColoc) by integrating GWAS summary statistics, QTL results, and linkage disequilibrium (LD) matrix data. GWAS summary statistics for birthweight were obtained from the Early Growth Genetics Consortium (https://egg-consortium.org/birth-weight-2019.html). We performed colocalization analysis on the sex-QTL loci separately for male- and female-stratified QTL, each integrated with GWAS summary statistics and LD matrix data from the 1000 G samples (Supplementary Data 13). A sex-QTL locus was tested for GWAS-colocalization provided a given GWAS summary statistics dataset (from 423 publicly available GWAS trait summary statistics available in ezQTL) contains a SNP within 1 Mb window from the QTL lead SNP with a GWAS association $P < 5 \times 10^{-8}$. 63 eQTL-GWAS locus pairs representing 42 unique phenotypes and 96 mQTL-GWAS locus pairs representing 16 unique phenotypes which fulfilled this requirement were tested, separately for males and females. Colocalization was considered significant if eCAVIAR colocalization posterior probability was $\geq 0.01$ or HyprColoc posterior probability was $\geq 0.5$ based on a window of 50 SNPs around the GWAS lead SNP, as recommended in ezQTL.

### Assessing phenotypic relevance

Associations of sex-DM CpGs or sex-DE transcripts with measures of neonatal anthropometry measured at birth (i.e., birthweight, birth length, head circumference) and placental size (i.e., placental weight and placental/birth weight ratio) were tested using the Spearman correlation test. The EWAS catalog (http://ewascatalog.org/)[110] and EWASAtlas (https://ngdc.cncb.ac.cn/ewas/atlas/index)[111] were searched to check whether the sex-DM and sex-mQTL CpGs overlap with previously known EWAS loci. The GWAS catalog v1.0.2 (https://www.ebi.ac.uk/gwas/; downloaded on December 2, 2021)[112] was searched to check whether the sex-eQTL SNPs, sex-mQTL SNPs, sex-DE genes, and sex-DM genes overlap with previously known GWAS loci.

### Functional annotation and pathway analysis

To test the enrichment of transcription factor binding sites (TFBS) in the QTL genetic variants, we separately submitted sex-eQTL and sex-mQTL SNPs to SNP2TFBS (https://epd.expasy.org/snp2tfbs/; a web interface for querying genetic variants that affect transcription binding sites)[113]. The sex-DM and sex-mQTL target CpGs were annotated for regulatory features from ENCODE, Roadmap Epigenomics, and GEN-CODE (https://zwdzwd.github.io/InfiniumAnnotation/EPIC_hm450_hg19.html). Enrichment of TFBS in the genomic region flanking 200 bp from sex-DM or sex-mQTL target CpGs was tested using the

JASPAR enrichment tool (https://jaspar.genereg.net/)[114]. JASPAR implements enrichment computations in the LOLA tool to test whether TFBS from the JASPAR database are enriched in our genomic regions of interest compared to CpG probes from the Illumina Infinium HumanMethylation450k microarray, separately for the following three sets of CpGs: i) CpGs hypermethylated in males, ii) CpGs hypermethylated in females, and iii) sex-mQTL targets CpG sites. We tested for enrichment of CpG positions with the functional elements DNase 1 hypersensitive sites (DHS), 15-state chromatin marks, and H3 histone marks compared to CpGs on the Illumina 450k array based on functional information from the Consolidated Roadmap Epigenomics database using eFORGE v2.0 (https://eforge.altiusinstitute.org/)[115].

Two sets of sex-DM genes (i.e., genes mapping male-hypermethylated CpG sites and female-hypermethylated CpGs) were separately annotated in biological context using the GENE2FUNC option using FUMA, a web-based platform that facilitates functional annotation of GWAS results (https://fuma.ctglab.nl/)[116]. Gene set enrichment analysis was performed using FUMA to test whether the input genes were enriched in GO Biological Process ontology and hallmark gene sets, which are biological states displaying coordinated expression as defined by the Molecular Signatures Database (MsigDB v7.0).

### Reporting summary
Further information on research design is available in the Nature Portfolio Reporting Summary linked to this article.

## Data availability
As part of the NICHD Fetal Growth Studies, which is the discovery cohort in the present study, the genotypes, DNA methylation, and gene expression data have been deposited in the dbGaP database under accession code phs001717.v1.p1 [https://www.ncbi.nlm.nih.gov/gap/?term=phs001717.v1.p1]. Moreover, as part of the Rhode Island Child Health Study (RICHS), which is the replication cohort in the present study, the genotypes and gene expression data have been deposited in the dbGaP database under accession code phs001586.v1.p1 [https://www.ncbi.nlm.nih.gov/projects/gap/cgi-bin/study.cgi?study_id=phs001586.v1.p1]. The following datasets made available by other groups were also used in the present analyses: the 1000 Genomes Reference Panel datasets were accessed at https://www.internationalgenome.org/category/reference/; the human genome reference made accessible by the Genome Reference Consortium was accessed at https://www.ncbi.nlm.nih.gov/assembly/GCF_000001405.13/; genotype imputation platform, as well as the Haplotype Reference Consortium reference panel, was accessed via the Michigan Imputation Server at https://imputationserver.sph.umich.edu/; and the Human Protein Altas dataset was downloaded at https://www.proteinatlas.org/about/download#protein_atlas_data. Source data are provided in this paper.

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

## Acknowledgements

This work was supported by the Intramural Research Program of the *Eunice Kennedy Shriver* National Institute of Child Health and Human Development (NICHD), National Institutes of Health (NIH) including American Recovery and Reinvestment Act funding via contract numbers HHSN275200800013C; HHSN275200800002I; HHSN27500006; HHSN275200800003IC; HHSN275200800014C; HHSN275200800012C; HHSN275200800028C; HHSN275201000009C and HHSN27500008. Additional support was obtained from the NIH Office of the Director, the National Institute on Minority Health and Health Disparities (NIMHD), and the National Institute of Diabetes and Digestive and Kidney Diseases (NIDDK). RICHS is partially supported by NIH-NIEHS R01ES022223 (C.J.M.), NIH-NIEHS R01ES022223-03S1 (C.J.M.), NIH-NICHD R01HD108310, and NIH-NIEHS U24ES028507 (C.J.M.). The authors acknowledge the research teams at all participating clinical centers for the NICHD Fetal Growth Studies, including Christina Care Health Systems, Columbia University, Fountain Valley Hospital, California, Long Beach Memorial Medical Center, New York Hospital, Queens, Northwestern University, University of Alabama at Birmingham, University of California, Irvine, Medical University of South Carolina, Saint Peters University Hospital, Tufts University, and Women and Infants Hospital of Rhode Island. Genotyping was performed in the Department of Laboratory Medicine and Pathology, University of Minnesota. The authors also acknowledge C-TASC and The EMMES Corporations in providing data and imaging support. The authors are also thankful to the RICHS study participants for their participation, and the study staff at Women and Infants Hospital for their dedication to the project. This work utilized the computational resources of the NIH HPC Biowulf cluster (http://hpc.nih.gov).

## Author contributions

F.T.-A. conceived and designed this study and wrote the draft manuscript. R.J.B. performed statistical analyses and visualizations. A.B., T.D.H., and P.W. contributed to data analysis. M.O. contributed to data analysis and write-up. C.J.M. and R.W. contributed data and samples. F.T.-A., R.J.B., A.B., T.D.H., P.W., C.J.M., M.O., and R.W. interpreted the results, reviewed the draft manuscript, provided critical intellectual content, and approved the final manuscript.

## Competing interests

The authors declare that they have no competing interests. Clinical trial registration: ClinicalTrials.gov, NCT00912132.
