## [Transparent Peer Review file · Nature Communications]

Sex-differentiated placental methylation and gene expression regulation has implications for neonatal traits and adult diseases

Corresponding Author: Dr Fasil Tekola-Ayele

Version 0:

Reviewer comments:

Reviewer #1

(Remarks to the Author)

This manuscript investigates sex differences in placental methylation (sex-DM) and expression (sex-DE), and in their genetic associations (sex-mQTL and sex-eQTL). The authors conducted comprehensive analyses and investigated the biological significance through integration and functional analyses. The manuscript is well-written, and the results are clearly presented. The observation of widespread sex differences in levels and genetic regulations of placental methylation is noteworthy. Additionally, the authors reported interesting findings such as female-biased methylation at KCND1 and IGF2R, suggesting an adaptive mechanism to enhance female fetal growth.

However, despite the significance of the scientific questions and the novelty of the study, there are some limitations. The study is hindered by small sample sizes (especially in expression data analysis), lack of rigorous replication, and many results requiring further validation. For instance, the majority of the identified sex-mQTLs and sex-eQTLs are low-frequency SNPs, raising concerns about false positives given the small sample size and the inclusion of many covariates. Below are some specific comments:

1. Supplementary Table 1: Please explain how the p-value was calculated and whether the comparison was stratified by race/ethnicity. Additionally, please include the duration of gestation if available.
2. Line 104: 'Methylation at sex-DM CpGs was more likely to be significantly correlated with nearby gene expression ...': Could this comparison (between sex-DM CpGs and non-sex-DM CpGs) be confounded by the detection of the sex-DM sites?
3. Supplementary Table 3: The majority of the 63 sex-DM CpGs appear to be positively correlated with nearby gene expression. Please explain the possible reasons behind this observation.
4. Line 107 and Figure 3: "The top eight strongest correlations..." – ZNF175 and NCAM2 were not among the "top eight" with the strongest correlations. Were they manually selected as examples for male or female hypermethylated DM CpGs negatively correlated with gene expression?
5. Lines 131-134: Please provide the allele frequency of the top SNPs (preferably separately for males and females) for easier interpretation of the results.
6. Figure 4: Please explain "Unclassified" in the figure legend.
7. Line 138: Provide the allele frequency for rs530705354.
8. Line 141: It should be rs35046330 according to Table S4.
9. Lines 145-147: Please describe the concordance of effect directions of the 19 "replicated" SNP-CpG pairs and indicate whether any of the top sex-mQTL described between lines 131-143 were replicated.
10. Line 165: Please describe the replication of sex-eQTL using RICHs.
11. Line 171 to 173: Is the methylation of cg25364822 correlated with expression level of FNDC5?
12. Figure 6. The numbers for target CpGs should be 521 (Line 10)?
13. Line 185: Please describe the concordance between the cellular composition estimated from methylation and from RNAseq data.
14. Line 193 to 199: This is an interesting observation. Please provide a more detailed description of the results, such as using Venn diagrams to show the overlap of positively correlated phenotypes in male and female hypermethylated sites, and describe several examples of the most significant sex-DM CpG associations with phenotypes.

15. Supplementary Table 9: Please add a column to show the significance of excessive overlapping.
16. Methods section: The regression models are not consistent between sex-DM and sex-mQTL, and between sex-DE and sex-mQTL analysis. For example, genotype PCs was not adjusted for in sex-mQTL and sex-eQTL analysis. Please discuss the impact of including many covariates in the association tests, especially considering the relatively small sample sizes.
17. Methods section: Is it possible to replicate sex-DM and sex-DE using the RICHs data?

Reviewer #2

(Remarks to the Author)

In this study, Tekola-Ayele et al. investigated the impact of sex on the methylome and transcriptome of human term placentae. A major finding was the strong effect of sex on placental methylome (with over 6,000 sex-DM CpGs) and its genetic regulation. In contrast, sex had a more subtle impact on placental transcriptome (with only 14 sex-DE genes) and its genetic regulation. The authors also report a strong enrichment of sex-DM at imprinted genes, some of which are imprinted only in placenta. In addition, the authors attempted to assess the phenotypic relevance of sex-biased placental methylation and gene expression by integrating their datasets with GWAS and EWAS catalogues, which led to the identification of several interesting co-localizations with birth weight and adult diseases. Overall, this is a well-conducted study, with good number of samples, robust analyses and validation of some parameters in an independent dataset. The paper is well written and the messages are clear. However, there are several points that I think will need to be addressed, which are presented below:

1. In the introduction, the authors claim that: "Whether sex-differentiated placental methylation is linked with gene expression regulation is unclear because no previous study has integrated the two molecular traits." This statement is not correct. A recent study by Czamara et al. (PMID: 38600394) published in April this year also performed genome-wide analyses of both DNA methylation and gene expression and reported a more pronounced impact of fetal sex on DNA methylation compared to gene expression. One of the strong points of the study by Czamara et al. is the inclusion of chorionic villus sampling collected from early placentae, which allowed them to look at stability of DNA methylation across two gestational time points. They reported more than 10 thousand sex-DMs, which may be due to the higher number of samples included (almost double compared to this study) and the newer methylation platform used (their EPIC arrays have 850K CpG compared to 450K CpGs in arrays used in this study). I think that the authors should remove the above statement from the introduction and use the findings by Czamara et al. to re-assess how many of the sex-DMs reported in this study are truly novel.
2. Although the findings reported in this manuscript are interesting, overall the paper remains descriptive, being heavily based on correlations. As per the famous phrase "correlation does not imply causation", I think we should attempt, whenever possible, to use ways to experimentally assess interesting data-driven hypotheses. We have now the ability to use human trophoblast cell culture with several lines being commercially available and having the sex-of-origin reported (e.g. BTS5, CT27 and CT30 are trophoblast stem cell lines of female origin and BT11 and CT29 trophoblast stem cell lines of male origin). As the authors acknowledge in the discussion, placenta cell composition is complex and the methods that assess cell composition in silico have considerable limitations. Although cell culture systems have their own limitations, the use of male and female human trophoblast cell lines would allow to independently validate sex-DM and sex-DE in a more homogeneous cell population. In addition, cell culture systems would create the opportunity to manipulate DNA methylation either at a global level (e.g. using inhibitors of DNMT, such as zebularine) or at specific loci, such as at those presented in Figure 3.
3. When reporting their findings regarding sex-mQTLs, the authors refer to some examples that are illustrated in Figure 4. However, in the panels of these figures we only see the location of the genetic variant (rs), but not that of the CpG (cg) or the gene that the text refers to. I think these panels should provide a better visual representation to match it with the text. A similar issue exists for the examples of sex-eQTLs illustrated in Figure 5: we see the position of the genetic variant (rs), but not that of the gene itself.
4. Related to the above point, when presenting sex-mQTLs, the authors report that the average distance between the genetic variant (rs) and the CpG (cg) is around 390 kb. However, there is no additional analysis or even a discussion on how a genetic variant could induce a sex-DM over such a long distance: could it be chromatin looping between the two loci or another mechanism? Regarding the first scenario, a good point of call would be the recent study by Varberg et al. 2023 (PMID: 37563143), which reported long-range chromatin interactions (by Hi-C) in the human trophoblast cell lines CT27 (female) and CT29 (male) in both stem cell status and upon differentiation to EVT (extravillous trophoblast). I think that exploring and integrating these datasets would give additional strength to the current study.
5. I understand the need to highlight some poster child examples in the abstract. However, I question the choice of KCNQ1 and IGF2R, for which the authors report in the abstract that their female-biased sex-DM "correlated with increased placental weight". However, from the results section, we learned that the correlation scores are very weak and for IGF2R the associated P value is not even significant if P<0.05 threshold is upheld (KCNQ1: cg13536051: rs=0.11, P=0.048 and cg15782852: rs=0.18, P=0.002; IGF2R: cg03634777: rs=0.11, P=0.064). There are probably several other specific examples in the study that could be highlighted instead.

Reviewer #3

(Remarks to the Author)

The authors used set out to assess differences in DNA methylation, gene expression and methylation- and expression-QTLs according to the sex of the human term placenta. Samples were collected as part of the study and underwent SNP genotyping with the HumanOmni2.5 Beadchip, DNA methylation profiling with the Infinium Methylation450 Beadchip, with a subset also undergoing RNA-seq. This study is important, since sex is so often overlooked in epigenetic analyses. In particular, male fetuses often cope much more poorly with pregnancy complications, the cause of which is unknown. Further,

differences between the sexes are present across the lifecourse, but which are dictated by life in utero is again unknown.

The paper is well written and clear, though I have several major comments regarding the gene expression correlation with sex-DM, and the section on genomic imprinting.

Major point 1: Correlation between sex-DM CpG methylation and gene expression – Line 99 – 119

I find this section to be underwhelming in its depth of thought and analysis. It is well established that the function of DNA methylation varies with context, whereby methylation close to the TSS impedes transcription, but in the gene body it is positively associated with transcription. As such, the text referring to whether the relationship between DNA methylation and gene expression is 'positive' or 'negative' is rather clunky, and would be improved with more information on precisely where the methylation in question is. Firstly, I suggest some additions to Figure 3 so that the spatial relationship between sex-DM CpGs and the genes (and their structure) suggested to be impacted by differential methylation can be seen visually. Currently, the location of methylation - 'body'/'TSS' is indicated only in a Supplementary Table.

Related to this, I could not find any information regarding the genomic distance of the differential methylation to the genes in question, only that they are considered to be 'nearby'. The only way to check, (unless the CpG is in the TSS, in which case this information is in Supp. Table 3, though I suggest moving to the main text for the highlighted genes) is for the reader to themselves input the CpG accession into the UCSC genome browser. Further, there is no information on other potentially relevant information on the CpGs, such as their position within predicted enhancers or insulators (though these elements can be very tissue-specific, this information remains useful). This could be included in the main text at least for the 8 genes highlighted in Figure 3. For instance, for the first CpG on the list of Supp. Table 3, (cg00874873), this region is predicted in USCS to be an enhancer, and the array probe is just slightly downstream of TFAP2 transcription factor binding sites, which are key in the placenta. A smaller point, but the 'strong' / 'weak' effect of differential methylation should also be put in the context of distance – i.e. distal differential methylation may have a weaker effect than the same methylation change close-by.

Major Point 2

The methods section indicates that only the closest gene to the CpG was considered: Line 435 "To evaluate relations between sex-DM CpGs and gene expression, we tested whether methylation at each sex-DM CpG is correlated with placental expression of its closest gene."

What about other nearby genes? Could you not repeat the analysis and look at genes within a defined window (100 kb?), this would then tell you whether the effect of differential methylation varies by distance. Methylation in a given sex-altered region may be having wide-ranging effects on gene expression, but this is missed in this analysis, and may explain why correlations between differential methylation and differential gene expression are low.

Potentially major point – though perhaps a wording error:

Line 104 – "Methylation at sex-DM CpGs was more likely to be significantly correlated with nearby gene expression in placenta at FDR-adjusted $P < 0.05$ than non-sex-DM CpGs located $>500\text{kb}$ away on the array (63/3429 vs. 395/59411; $\chi^2 = 61.6$, $P = 4.21 \times 10^{-15}$)."

I'm confused by the wording here, else the premise for this comparison is flawed – are you saying that methylation at sex-DM CpGs was more likely to be correlated with nearby gene expression than non-sex DM CpGs (reasonable) or compared to non-sex DM CpGs that are also $>500\text{ kb}$ away – this latter option is not a good comparison since you are not separating out the fact of their sex-DM or non-sex-DM status from the distance!

Minor points:

Line 110 – you need to cite here PMID: 29376485 Gong et al 2018 (currently reference 19 in your manuscript) since they already found the sex-specific placental DMR at CSMD1 and characterised it extensively.

Whether sex-differentiated placental methylation is linked with gene expression regulation is unclear because no previous study has integrated the two molecular traits – this is not entirely true – your reference PMID: 28234023 a study on sexual dimorphism do integrate DNAm and expression, though they use gene expression data from a previous published study, and again PMID: 29376485 also link placenta methylation and gene expression

Line 137 – "7.9% (134/1701) of sex-mQTLs had concordant effects in males and females", ok so what makes them sex-mQTL if they are concordant in males and females? Is it the level of DNAm? Please explain in the text.

Enrichment for Imprinted Genes, Line 228 onwards

The descriptions of imprinting are, in general, imprecise. Namely, whilst IGF2R is a powerful regulator of placental growth, evidence of its imprinting in human is weak, whereby it is polymorphic/rare. As such, the use of IGF2R by the authors as an exemplar imprinted gene in humans involved in the conflict hypothesis is not appropriate. Further, IGF2R is not involved in Beckwith Wiedemann Syndrome.

Line 237 – "such as insulin-like growth factor type 2 receptor (IGF2R) and potassium voltage-gated channel subfamily Q member 1 (KCNQ1), which are paternally imprinted (maternally expressed) in placenta and involved in regulation of fetoplacental growth and development ref52, 53"

Reference 52 is a purely mouse paper, with no analysis of IGF2R in humans. Human IGF2R imprinting, whilst it is maternally expressed, is highly polymorphic in and reported in only a minority of individuals (PMID: 16614068). Loss of imprinting of this gene has also not been linked with any human pathology. There is also no strong evidence that the KCNQ1 gene is involved in fetoplacental growth/development, except of the heart (the wider locus notwithstanding)

Line 322, regarding sex biased CpG methylation “This observation is aligned to the role of maternally expressed imprinted genes, as opposed to paternally expressed ones, in limiting fetal growth to conserve maternal resources”

And:

Line 329 “Our observation that epigenetic processes at KCNQ1 and IGF2R that promote fetoplacental growth were female-biased is not compatible with the well-known male-bias in fetoplacental growth, and raises a potential crosstalk between sex and imprinting regulation in placenta. The male placenta functions near its maximum capacity prioritizing fetal growth, whereas the female placenta prioritizes maternal reserve”

Aside from the fact that, as discussed above, KCNQ1 and IGF2R are not, in humans, strong candidates for exemplar ‘conflict hypothesis’ genes, I believe the rationale here is not correct. I do not agree that the male placenta prioritises fetal growth and the female placenta prioritises maternal reserve. All placentas are selfish. Instead, females, differently to males, reduce growth to preserve other fetal reserves in case of maternal stress (such as asthma or pre-eclampsia) whereas males increase growth to try and improve their chances. Different strategies, but both designed to improve fetal chances of survival, regardless of maternal reserve.

Line 324 “repressed expression of KCNQ1 and IGF2R cause Beckwith-Wiedemann syndrome”

There is no evidence that dysregulated imprinting of IGF2R causes BWS (do the authors mean IGF2?). Further, whilst the KvDMR is highly implicated in BWS pathogenesis, the KCNQ1 gene itself is not – CDKN1C is a much more important gene here. Did you check the expression of other genes in the locus?

Line 333: “We posit that there may be sex-dependent epigenetic regulation of imprinting at KCNQ1 and IGF2R as a compensatory adaptive mechanism to counter-balance resource allocations to the fetus to prevent undergrowth in females and overgrowth in males”. This is unlikely - there are no sex biases in the clinical disorders caused by LOI at KCNQ1, and by necessity the germline KvDMR imprint is very stable.

Minor Comments:

Line 347: “Third, sex is correlated with numerous environmental factors, which makes it difficult to dissociate differences due to biological sex and other factors correlated with biological sex”

I entirely disagree here – a unique strength of this study is that you are looking at the placenta, and therefore at sex differences in utero. This is prior to the manifestation of the vast environmental differences faced by ex-utero males and females. Prenatally, the environment the mother provides for her male or female fetus, aside from innate fetal hormonal and developmental differences (which I do not believe the authors mean when they say ‘environmental’), is likely to be very similar regardless of sex, and so in fact not subject to environmental differences.

Line 150: “None of the 14 genes exhibited sex difference in non-placental tissues from the genotype tissue expression (GTEx) portal, aligning with evidence of poor correlation of sex-differentiated autosomal gene expression between term placenta and 42 non-reproductive adult tissues from GTEx”

-This is not surprising since the cited report finds sex specific differences overall from GTEx to be very tissue specific (as the authors note on line 164). However, it is not clear whether the placenta has a uniquely poor correlation with other tissues? (i.e. compared to say, Liver vs Brain or Lung?) – it would be interesting to expand slightly on this with more computational analysis, to establish whether placenta is an outlier (as is often the case in terms of tissue specific gene regulatory profiles.)

Line 239, plus abstract and various other places: Please don’t use the term imprinted to mean the silencing of an imprinted gene - say silenced, or repressed allele.

Regarding the discussion of the overrepresentation of imprinted genes in sex-biases – rather than focusing on resource allocation – perhaps the authors could speculate instead about whether particular epigenetic facets of imprinted domains could leave them vulnerable to sex-specific alterations?

Line 269: Enrichment for TFs such as hormone receptors - perhaps this is not a surprising finding in the context of sex differences, NR2C2 is involved in spermatogenesis – could the authors comment further on this?

Line 282: “methylation sites that were sex-biased were not enriched for TF binding” this should say ‘predicted’ TF binding, or instead ‘TF binding sites’.

The Supplementary table PDF is not complete and appears to only have table 1

Version 1:

Reviewer comments:

Reviewer #1

(Remarks to the Author)

In this revised version, the authors have addressed my review questions, added new analyses, and I have no further comments. Just one side note: Ref 68 has been published in Nature Genetics, 55:1807–1819 (2023), and this reference also showed a maternal-only association between a variant in intron 10 of KCNQ1 and placenta weight.

Reviewer #2

(Remarks to the Author)

In the revised manuscript, Tekola-Ayele et al. addressed many of the points raised on the original manuscript by myself and my colleagues who reviewed the paper. Although they did not perform any additional experimental work to address the functional relevance of their findings (point 2 in my previous comments), the analyses performed are welcome additions to the paper. On the flip side of the coin, although there are some differences from the work performed by Czamara et al. (PMID: 38600394) regarding the approaches used to analyse their data (these differences are mentioned in the reply to reviewer's comments), in my view the main messages of the two papers are highly similar:

- sex has a strong impact on placental DNA methylation, with most sex-DM being hypermethylated in males;
- differences in placental gene expression between the two sexes are limited;
- there are very few loci for which there is a direct triad relationship methylation-expression-phenotype.

The example on the imprinted cluster located on chromosome 11p15.5 remains rather underwhelming, because most of the correlations listed are weak. There is no explanation for why this cluster of imprinted genes may be more "susceptible" to acquiring sex-DM compared to other imprinted loci.

Because the correlations between sex-DM and phenotypes do not seem to depend in most cases by changes in gene expression, the authors should explore more their value as biomarkers of disease. Instead of focussing on correlations between individual CpGs and phenotypes, could the authors try to identify combinations of these CpG that have strong predictive value of adverse pregnancy outcomes? Something along the way DNA methylation clock was developed, with several hundred of CpGs being able to accurately predict age.

As a minor point, the authors should check their list of references. After the edits made, some papers do not seem to be cited any longer in the text.

Reviewer #3

(Remarks to the Author)

I find the manuscript to be much improved and the authors carefully answered all my previous points. My remaining comments below are largely stylistic, though there are still a couple of areas requiring a bit more precision regarding descriptions of imprinting. I think a couple of edits will have been lost to formatting - please also see pdf file attached here.

Abstract:

Line 33: "...known KCNQ1OT1/CDKN1C imprinting cluster of maternally expressed genes" – they are not all maternally expressed. The ncRNA (and locus-controlling transcript) KCNQ1OT1 is paternally expressed.

Line 61 – nearby genomic (no s) region

Line 102: "Differentially methylated sites may be involved in transcript regulation depending on context.

Line 103: "Therefore, we assessed whether the sex-differentially methylated sites are correlated with nearby gene expression more than expected by chance.

These are two separate points. The first could be described more clearly – suggest instead "Differentially methylated sites may be involved in transcript regulation, but their precise effect is likely to depend on the genomic context".

Regarding the second point, whether sex-DMs are correlated with gene expression more than expected by chance, is asking whether there is some selection for sex-DMs to regulate genes. Suggest moving lines 103-108 'Therefore we assessed....against all genes within 200kb distance' down, and addressing your context-dependent analysis first, following line 102 directly with the end of Line 108 – "Positive sex-DM CpG-gene correlations...."

It is worth clarifying in the text what you mean by positive sex-DM CpG-gene correlations – I suppose it is that the methylated CpG state correlates with increased gene expression?

Line 112: "These findings align with studies that showed that methylation within gene body is often positively correlated with gene expression while methylation in TSS is often negatively correlated with gene expression"

More of a stylistic issue here but the description used by the authors here "findings align with studies that showed" is not quite appropriate – the relationship between gene-body methylation and transcription has been well established since 1984 and should be described as such. Something like: "These findings are consistent with the common theory that methylation within the gene body is positively correlated with gene expression and methylation in TSS is negatively correlated with gene expression" (insert key references such as Jones, P. A., 1999 PMID: 10087932, or Jones P.A., 2012 PMID: 22641018, Laurent et al., 2010 PMID: 20133333).

Line 114 "Positive correlation between methylation and gene expression may be explained by mechanisms such as preferential binding of transcription factors to highly methylated sites and binding of gene-repressive transcription factors to unmethylated sequences." I don't think this sentence needs to be there, it is a bit misleading and detracts from the main point.

I suggest inserting the lines 103-108 'Therefore we assessed....against all genes within 200kb distance' from above into Line 117: "We then assessed whether the sex-differentially methylated sites are correlated with nearby gene expression more than expected by chance. Methylation at sex-DM CpGs was more likely to be significantly correlated with nearby gene expression in placenta at FDR-adjusted $P < 0.05$ than non-sex-DM CpGs on the array (63/3636 vs. 1288/277343; $\chi^2 = 120.6$, $P = 4.67 \times 10^{-28}$). An additional 13 CpG-gene correlations were identified by testing the correlation of sex-DM CpGs against all genes within 200 kb distance.

I think you need to follow this with some explanation - Why did you do this calculation? - What do the authors think the significance of their finding that sex-DMs correlate with gene expression more highly than that expected by chance? (when

comparing to all other CpGs on the array). This is not discussed, but is a key point.

The end of this section is great.

Line 146, small errors/grammar: "...suggesting that genetic variants anywhere within the 1Mb distance can have sex-dependent effects on methylation in placenta."

Line 157: Across the 1839 sex-mQTLs, the genetic variant's distance from its associated CpG was inversely correlated with its sex-biased effect on methylation ($r_s = -0.20$, $P = 7.32 \times 10^{-19}$). Genetic variants may induce epigenetic regulation in placenta through long-range chromatin interactions³⁵

To make this clearer I suggest the following change:

Across the 1839 sex-mQTLs, the genetic variant's distance from its associated CpG was inversely correlated with its sex-biased effect on methylation ($r_s = -0.20$, $P = 7.32 \times 10^{-19}$) which was to be expected, since previous evidence indicates that >75 % gene-regulatory-element interactions are short range (Fuentes et al., ELife 2018, PMID: 30070637). However, genetic variants may induce epigenetic regulation in placenta through long-range chromatin interactions³⁵ and distal transcript regulatory elements such as enhancers.^{41,42}

Line 304 "...genes such as KCNQ1 that exhibit biallelic expression". This is not quite correct – the cited paper demonstrates that KCNQ1 is imprinted in fetal placenta, (and it is shown elsewhere to be imprinted in many other tissues). You could say 'in term placenta', but in this case you need to say that KCNQ1OT1 is also biallelic at term, as Monk et al., show in the cited paper.

Line 317 – 'imprinting dysregulation...' rather implies that your observations are akin to imprinting being disrupted according to sex, however, I don't think this is the case - KCNQ1OT1 is paternally expressed, whilst CDKN1C is maternally expressed, so a mix of paternally and maternally expressed genes at the same locus exhibit upregulation in the presence of maternal hypermethylation. To me, this indicates that this is unrelated to imprinting, or at least it is not a strong pattern. It is fine to mention imprinting dysregulation and links to disease, but perhaps make clear that this is simply another way that dosage of these genes is altered.

Also, you mention SLC22A18, but according to Supp Table (and the positions of the Sex-DM CpGs) they are actually in SLC22A18AS.

From Supplementary Table 11: 29 of the Sex-DM CpGs are in TP73 – perhaps the authors could comment on this gene, there is a large enrichment here, and also in HOXA4 (19 here). Also, the 8 Sex-DM CpGs in KCNQ1 are listed twice (due to the gene being derived from both otago and geneimprint lists) giving a false impression of the number of CpGs present in this gene.

Lines 401-416: I find the summary and potential reasoning for imprinting sex bias to be much improved.

Line 404 - In our data, higher methylation of the female-hypermethylated CpGs was associated with female-specific upregulation or male-specific downregulation of maternally expressed genes in this imprinting domain. These findings suggest a potential crosstalk between sex and imprinting regulation in placenta."

Related to my query above, and now misstated here in the discussion, the authors mention 'female-specific/male-specific up/down regulation of maternally expressed genes' - KCNQ1OT1 is one of the genes whose expression increases in response to increased CpG methylation in maternal placentas, but it is paternally expressed, which again calls into question the rationale regarding sex and 'imprinting cross-talk'. I feel the following discussion by the authors on unique epigenetic architecture to be a much more persuasive argument towards the apparent enrichment of imprinted genes here.

RESPONSE TO REVIEWER COMMENTS

Reviewer #1 (Remarks to the Author):

This manuscript investigates sex differences in placental methylation (sex-DM) and expression (sex-DE), and in their genetic associations (sex-mQTL and sex-eQTL). The authors conducted comprehensive analyses and investigated the biological significance through integration and functional analyses. The manuscript is well-written, and the results are clearly presented. The observation of widespread sex differences in levels and genetic regulations of placental methylation is noteworthy. Additionally, the authors reported interesting findings such as female-biased methylation at KCND1 and IGF2R, suggesting an adaptive mechanism to enhance female fetal growth.

Response: Thank you for your thoughtful consideration of the manuscript and positive remark. The constructive feedback was helpful to improve the manuscript.

However, despite the significance of the scientific questions and the novelty of the study, there are some limitations. The study is hindered by small sample sizes (especially in expression data analysis), lack of rigorous replication, and many results requiring further validation. For instance, the majority of the identified sex-mQTLs and sex-eQTLs are low-frequency SNPs, raising concerns about false positives given the small sample size and the inclusion of many covariates. Below are some specific comments:

Response: The sample size of the RNA sequencing dataset is smaller than that of the methylation dataset, and we expanded on our discussion on this in the Discussion section (lines 465-474).

“Fourth, detection of sex-eQTLs may have been limited by the sample size of our placental RNA-seq dataset. The inclusion of covariates to account for genetic population structure and cell heterogeneity may have further impacted the study’s power in detecting weaker signals. Similar to our findings, limited sex difference in gene expression regulation has been observed in 44 non-placental tissues from GTEx⁴⁶ and in adult blood.^{98,99} Moreover, one study demonstrated that samples of millions of individuals would be required to detect sex-biased eQTLs in blood that appreciably mediate sex-biased genetic associations with complex traits.⁹⁸ Larger datasets and accounting for factors that influence QTLs can improve the transferability of sex-QTLs found to be limited in our study as well as others.^{46,100}”

We note that our RNA-seq dataset’s sample size is comparable with GTEx’s kidney dataset reported in their sex-biased eQTL study (Oliva et al Science. 2020; 369(6509): eaba3066). In addition, our cohort is among few placenta cohorts with multi-omics profiles, addressing an important gap in the field. With suggestion (including by reviewer #2), we have performed additional look up of our sex-DM, sex-DEG, and sex-mQTL in Czamara et al.’s work and performed additional evaluation of the sex-DEGs in the RICHS cohort, finding additional replication of our sex-DM, sex-DEG, and sex-mQTL associations. We updated the manuscript on those findings (lines 102, 186-189, 191-

194). All identified sex-eQTL SNPs are “common-frequency” variants (i.e., have MAF >5%; none of which is “low-frequency” with MAF <5%), with mean(\pm sd) of 8.9% (\pm 4.2%) for sex-mQTL and 6.3% (\pm 1.0%) for sex-eQTLs (Table S4 column H and Table S7 column I list each QTL SNP’s allele frequency).

1. Supplementary Table 1: Please explain how the p-value was calculated and whether the comparison was stratified by race/ethnicity. Additionally, please include the duration of gestation if available.

Response: We have added the following on p-value calculation methods to Supplementary Table 1 footnote: “** t-test was applied for continuous variables, and chi-squared test was applied for categorical variables”. Duration of gestation is already included in the table (see the variable “gestational age at delivery” in weeks). The comparison was not stratified by race/ethnicity because that is not the focus of the paper.

2. Line 104: ‘Methylation at sex-DM CpGs was more likely to be significantly correlated with nearby gene expression ...’: Could this comparison (between sex-DM CpGs and non-sex-DM CpGs) be confounded by the detection of the sex-DM sites?

Response: We think the sentence in line 104 was confusing because the rationale was not clarified. Our intention is to assess whether sex-differentially methylated sites have more functional relevance in transcript silencing or activation than what is expected by chance (i.e., compared to other methylation sites on the array). As expected, sex-DMs were more likely to be correlated with gene expression, suggesting that the sex-DM CpGs are enriched for genome regions involved in transcript regulation. We have now added the following sentences to clarify the rationale (lines 106-108):

“Differentially methylated sites may be involved in transcript regulation depending on context.³⁰ Therefore, we assessed whether the sex-differentially methylated sites are correlated with nearby gene expression more than expected by chance.”

We have also updated the comparison to be between sex-DM CpGs and non-sex-DM CpGs (irrespective of distance) as suggested by reviewer #2 (lines 110-112). The conclusions remain the same.

3. Supplementary Table 3: The majority of the 63 sex-DM CpGs appear to be positively correlated with nearby gene expression. Please explain the possible reasons behind this observation.

Response: Thank you for this suggestion. We have added the following sentences (lines 112-121).

“Positive sex-DM CpG-gene correlations were more common than negative correlations. Moreover, the majority of CpGs positively correlated with gene expression were in the gene body, whereas the majority of those with negative correlation were in the gene transcription start site (TSS) (**Supplementary Table 3**). These findings align

with studies that showed that methylation within gene body is often positively correlated with gene expression while methylation in TSS is often negatively correlated with gene expression.^{30,31} Positive correlation between methylation and gene expression may be explained by mechanisms such as preferential binding of transcription factors to highly methylated sites³², and binding of gene-repressive transcription factors to unmethylated sequences.³³

4. Line 107 and Figure 3: “The top eight strongest correlations...” – ZNF175 and NCAM2 were not among the “top eight” with the strongest correlations. Were they manually selected as examples for male or female hypermethylated DM CpGs negatively correlated with gene expression?

Response: We have expanded the figure legend to clarify that the top 8 genes were selected based on strength of correlation coefficient, consisting of the two strongest male-hypermethylated positive correlations, two strongest male-hypermethylated negative correlations, two strongest female-hypermethylated positive correlations, and two strongest female-hypermethylated negative correlations. Note: in response to a comment by reviewer #3, we have added regulatory annotations in the figure, so we have split Figure 3 into two separate sex-specific figures (Figure 3 for four male-hypermethylated and Figure 4 for four female-hypermethylated associations), and the manuscript text has been updated to match this update.

Figure 3. Top four male-hypermethylated DNA methylation sites significantly correlated with nearby gene expression in placenta and overlap with regulatory regions. Top associations were selected based on strength of positive and negative correlations, two from each. The displayed figures include male-hypermethylated CpGs that showed the top two strongest negative correlations (**A, B**) and the top two strongest positive correlations (**C, D**) with their corresponding nearby genes and UCSC Genome Browser plots of regulatory sites. For each CpG, the leftmost plot shows correlation with gene expression in placenta (Spearman correlation (ρ) with FDR-adjusted $P < 0.05$ was considered significant); the middle plot shows UCSC Genome Browser’s regulatory sites from ENCODE; the rightmost plot shows UCSC Genome Browser’s regulatory sites in placenta smooth chorion. Red highlights indicate the gene TSS; blue highlights indicate the CpG; and purple highlights indicate that the gene TSS and CpG are in close proximity to each other. Tracks labeled “H3K4me1”, “H3K4me3”, and “H3K27ac” represent H3K4me1, H3K4me3, and H3K27ac marks found in 7 cell lines from ENCODE and in placenta smooth chorion. Tracks labeled “DNase1” represent DNaseI hypersensitivity clusters in 125 cell types from ENCODE. Tracks labeled “ChIP-seq” represent ChIP-seq clusters in 338 factors and 130 cell types from ENCODE and ChIP-seq marks found in placenta smooth chorion. Tracks labeled “Promoter/enhancer” and “Interaction” represent GeneHancer Double Elite Regulatory Elements and GeneHancer Double Elite Clustered Interactions, respectively. Tracks labeled “H3K27me3” and “H3K36me3” represent H3K27me3 and H3K36me marks in placenta smooth chorion, respectively.

Figure 4. Top four female-hypermethylated DNA methylation sites significantly correlated with nearby gene expression in placenta and overlap with regulatory regions. Top associations were selected based on strength of positive and negative correlations, two from each. The displayed figures include female-hypermethylated CpGs that showed the top two strongest negative correlations (**A, B**) and the top two strongest positive correlations (**C, D**) with their corresponding nearby genes and UCSC Genome Browser plots of regulatory sites. For each CpG, the leftmost plot shows correlation with gene expression in placenta (Spearman correlation (ρ) with FDR-adjusted $P < 0.05$ was considered significant); the middle plot shows UCSC Genome Browser's regulatory sites from ENCODE; the rightmost plot shows UCSC Genome Browser's regulatory sites in placenta smooth chorion. Red highlights indicate the gene TSS; blue highlights indicate the CpG; and purple highlights indicate that the gene TSS and CpG are in close proximity to each other. Tracks labeled "H3K4me1", "H3K4me3", and "H3K27ac" represent H3K4me1, H3K4me3, and H3K27ac marks found in 7 cell lines from ENCODE and in placenta smooth chorion. Tracks labeled "DNase1" represent DNaseI hypersensitivity clusters in 125 cell types from ENCODE. Tracks labeled "ChIP-seq" represent ChIP-seq clusters in 338 factors and 130 cell types from ENCODE and ChIP-seq marks found in placenta smooth chorion. Tracks labeled "Promoter/enhancer" and "Interaction" represent GeneHancer Double Elite Regulatory Elements and GeneHancer Double Elite Clustered Interactions, respectively. Tracks labeled "H3K27me3" and "H3K36me3" represent H3K27me3 and H3K36me marks in placenta smooth chorion, respectively.

5. Lines 131-134: Please provide the allele frequency of the top SNPs (preferably separately for males and females) for easier interpretation of the results.

Response: MAF has now been added for all QTLs in the text. The correlation in MAF between males and females is very high ($r=0.96$, $p < 2.2e-16$), so we did not report these separately by sex.

6. Figure 4: Please explain "Unclassified" in the figure legend.

Response: We have added the following sentence to the Figure 5 legend: "Unclassified includes associations that do not fall into the other categories (i.e., sex-biased mQTL associations with FDR-adjusted $P \geq 0.05$ in both sex strata and/or those with FDR-adjusted $P_{diff} \geq 0.05$).

7. Line 138: Provide the allele frequency for rs530705354.

Response: We have added the MAF, which is 6.2%.

8. Line 141: It should be rs35046330 according to Table S4.

Response: Thank you. The SNP ID is now corrected as "rs71304466"

9. Lines 145-147: Please describe the concordance of effect directions of the 19 “replicated” SNP-CpG pairs and indicate whether any of the top sex-mQTL described between lines 131-143 were replicated.

Response: Thank you. We have now added the following sentences (lines 185-189):

“...19 were significant at FDR-adjusted $P < 0.05$ and with consistent effect direction. The strongest male-specific mQTL association identified (i.e., rs12242275/rs7093024-cg09082518 [CCDC6]) was among those replicated in the RICHS dataset (Supplementary Table 5). One of our sex-mQTLs (rs34571066-cg13299927 [ARHGEF10L]) has previously been reported.²⁰”

10. Line 165: Please describe the replication of sex-eQTL using RICHS.

Response: This has been described in the Methods section, and we have now added the following sentence in the Results section (lines 212-214):

“Out of the 13 sex-eQTLs, only rs10892219-*ATP5MG* was available in the RICHS dataset and did not exhibit significant association after FDR-adjustment.”

11. Line 171 to 173: Is the methylation of cg25364822 correlated with expression level of *FNDC5*?

Response: Thank you for this helpful comment. Yes, cg25364822 and *FNDC5* were correlated in males, consistent with the sex-DM and sex-DE findings. We have now added the following sentence, also expanding the potential impact of the locus in placental function and pregnancy complications (lines 222-230):

“In turn, higher methylation at cg25364822 was significantly correlated with lower *FNDC5* expression in males ($r_s = -0.36$, $P = 0.023$), but not in females ($r_s = 0.02$, $P = 0.91$). *FNDC5* is a precursor of irisin which protects the placenta from oxidative stress, dysregulation of placental trophoblast differentiation, and insulin resistance.^{52,53} Lower irisin levels have been linked to preeclampsia^{52,53}, a placenta insufficiency-disorder correlated with smaller placental weight.⁵⁴ In our data, increased cg25364822 methylation and decreased *FNDC5* expression were correlated with smaller placental weight ($r_s = -0.12$, $P = 0.04$; $r_s = 0.29$, $P = 0.01$, respectively), suggesting a sex-biased epigenetic process that may regulate gene expression levels that impact placental function.”

12. Figure 6. The numbers for target CpGs should be 521 (Line 10)?

Response: Thank you for catching this oversight. This is now corrected.

13. Line 185: Please describe the concordance between the cellular composition estimated from methylation and from RNAseq data.

Response: Our goal in using both approaches is to cross-validate whether our results are explained by cell composition estimates based on different approaches. The cell composition estimation method that uses methylation and the method that uses RNA-seq data produce cell types that differ in number and type. Whereas the methylation-based estimator calculates proportion of 6 cell types, the RNA-seq-based estimator calculates 27 cell types that include fetal and maternal cells and cellular sub-types, with few overlaps. Therefore, it is impossible to do a one-to-one comparison of concordance. More work is needed in this area for placenta.

14. Line 193 to 199: This is an interesting observation. Please provide a more detailed description of the results, such as using Venn diagrams to show the overlap of positively correlated phenotypes in male and female hypermethylated sites, and describe several examples of the most significant sex-DM CpG associations with phenotypes.

Response: Thank you. We have added a new Figure to illustrate this better (Figure 8, an upset plot showing sex-DM CpGs with overlapping correlation with neonatal and placental measures). We have also added the following sentences to highlight important loci (lines 256-269).

“Among male-hypermethylated CpGs, ten were positively correlated with all three neonatal anthropometry measures and negatively correlated with placenta-birthweight ratio (CpGs map to *EGLN1*, *FNBP1L*, *TSPAN14*, *TEAD1*, *MIR130A*, *TMEM223*, *DOCK9*, *SLC1A5*, *DGCR6L*, and *ST6GALC6* genes). *EGLN1* encodes a protein in the hypoxia-inducible factor (HIF) pathway, and is important for placental development through its role as oxygen sensor and regulator of HIF1A expression in trophoblast cells.^{58,59} Among female-hypermethylated CpGs, three were positively correlated with placenta-birth weight ratio and negatively correlated with at least two neonatal anthropometry measures (CpGs map to *SHANK3*, *POP4*, and *GRHL3* genes). *SHANK3* encodes a synaptic protein that is crucial for the development of brain circuits.⁶⁰ The *SHANK3* gene locus has been implicated in cognitive function measurement and schizophrenia^{61,62}, and in autism spectrum disorders⁶⁰ which is more commonly diagnosed in males than females.⁶³ Placental hypomethylation at a nearby locus (22q13.33) has been linked with autism⁶⁴, and hypomethylation of our female-hypermethylated CpG in adult blood DNA has been associated with brain gray matter volume.⁶⁵”

15. Supplementary Table 9: Please add a column to show the significance of excessive overlapping.

Response: We have added a column for p-values and FDR-adjusted P-values of hypergeometric test of enrichment in Supplementary Table 9. Several trait associations exhibited significant enrichment.

16. Methods section: The regression models are not consistent between sex-DM and sex-mQTL, and between sex-DE and sex-mQTL analysis. For example, genotype PCs was not adjusted for in sex-mQTL and sex-eQTL analysis. Please discuss the impact of including many covariates in the association tests, especially considering the relatively small sample sizes.

Response: Thank you for bringing this point. We apologize for the error in reporting the covariates in the QTL models - genotype PCs have been included in sex-mQTL and sex-eQTL analysis, but have been inadvertently left out in the manuscript text. We have now corrected those sentences (line 569-570, 602, 610). In the sex-stratified follow-up analyses, we reduced the number of PEER factors and genotype PCs to accommodate the smaller sample sizes in the sex-stratified data. We expanded the Discussion section reflecting this (line 465-474):

“Fourth, detection of sex-eQTLs may have been limited by the sample size of our placental RNA-seq dataset. The inclusion of covariates to account for genetic population structure and cell heterogeneity may have further impacted the study’s power in detecting weaker signals. Similar to our findings, limited sex difference in gene expression regulation has been observed in 44 non-placental tissues from GTEx⁴⁶ and in adult blood.^{98,99} Moreover, one study demonstrated that samples of millions of individuals would be required to detect sex-biased eQTLs in blood that appreciably mediate sex-biased genetic associations with complex traits.⁹⁸ Larger datasets and accounting for factors that influence QTLs can improve the transferability of sex-QTLs found to be limited in our study as well as others.^{46,100}”

17. Methods section: Is it possible to replicate sex-DM and sex-DE using the RICHS data?

Response: Thank you for this suggestion. The RICHS dataset has been included in a previous sex-DM meta-analysis study, which has already been included in the retrospective look up of our sex-DM results as presented in Supplementary Table 2 and Results text. We have now tested replication of the sex-DE in RICHS, finding that 10 out of 12 DEGs that are available in RICHS showed directionally consistent sex-DE associations, but were not significant after FDR-adjustment.

The following sentences have been added to the Methods section (line 629-630, 633-634):

“Of the 14 sex-DE associations discovered, 12 transcripts with RNA-seq data in RICHS were tested.”

“Sex-DE analyses were performed using linear regression models as in the discovery analysis with adjustment for race/ethnicity, gestational age at delivery, and 10 genotype PCs.”

The following sentences have been added to the Results section (line 191-194):

“We found 14 sex-DE (FDR-adjusted $P < 0.05$), of which sex-DE of *ANGPT2* has been previously reported²⁰ and 13 were novel (**Figure 2, C; Supplementary Table 6**). In the RICHHS dataset, 12 sex-DE transcripts were available, and 10 showed directionally consistent sex-DE associations, but none was significant after FDR-adjustment (**Supplementary Table 6**).”

Reviewer #2 (Remarks to the Author):

In this study, Tekola-Ayele et al. investigated the impact of sex on the methylome and transcriptome of human term placentae. A major finding was the strong effect of sex on placental methylome (with over 6,000 sex-DM CpGs) and its genetic regulation. In contrast, sex had a more subtle impact on placental transcriptome (with only 14 sex-DE genes) and its genetic regulation. The authors also report a strong enrichment of sex-DM at imprinted genes, some of which are imprinted only in placenta. In addition, the authors attempted to assess the phenotypic relevance of sex-biased placental methylation and gene expression by integrating their datasets with GWAS and EWAS catalogues, which led to the identification of several interesting co-localizations with birth weight and adult diseases. Overall, this is a well-conducted study, with good number of samples, robust analyses and validation of some parameters in an independent dataset. The paper is well written and the messages are clear. However, there are several points that I think will need to be addressed, which are presented below:

Response: We appreciate the reviewer's positive remark and insightful comments, which gave us an opportunity to strengthen the validity of the findings and clarify the manuscript.

1. In the introduction, the authors claim that: "Whether sex-differentiated placental methylation is linked with gene expression regulation is unclear because no previous study has integrated the two molecular traits." This statement is not correct. A recent study by Czamara et al. (PMID: 38600394) published in April this year also performed genome-wide analyses of both DNA methylation and gene expression and reported a more pronounced impact of fetal sex on DNA methylation compared to gene expression. One of the strong points of the study by Czamara et al. is the inclusion of chorionic villus sampling collected from early placentae, which allowed them to look at stability of DNA methylation across two gestational time points. They reported more than 10 thousand sex-DMs, which may be due to the higher number of samples included (almost double compared to this study) and the newer methylation platform used (their EPIC arrays have 850K CpG compared to 450K CpGs in arrays used in this study). I think that the authors should remove the above statement from the introduction and use the findings by Czamara et al. to re-assess how many of the sex-DMs reported in this study are truly novel.

Response: We thank the reviewer for bringing our attention to this work that was published during submission of our manuscript. We have revised the sentence (line 64), removing "...because no previous study has integrated the two molecular traits". We have also added this work to the list of prior studies on sex-biased methylation and gene expression in Supplementary Table 2. We found that 1483 sex-DM overlapped, of which 1149 have been found by previous studies listed in Table S2 and 334 were new, so the percent sex-DM that are novel is now 41.1% (previously 47%). We have also updated the text (line 101-103) as follows:

“Out of the 6,077 sex-DM, 41.1% (2497/6077) were novel and the remaining 58.9% have previously been reported with consistent effect direction^{18-20,27-29} (**Figure 2, A; Supplementary Table 2**).”

One sex-DE overlapped with ours, so we have updated the text (line 191-192) as follows:

“We found 14 sex-DE (FDR-adjusted $P < 0.05$), of which sex-DE of *ANGPT2* has been previously reported²⁰ and 13 were novel (**Figure 2, C; Supplementary Table 6**).”

One sex-mQTL overlapped, so we have updated the text (line 188-189) as follows:

“One of our sex-mQTLs (rs34571066-cg13299927 [*ARHGEF10L*]) has previously been reported.²⁰”

We would like to note important differences between our study and Czamara et al's. 1) Sex-eQTLs were not assessed by Czamara et al. 2) Whether sex-DM was associated with nearby gene expression in placenta was not tested by Czamara et al. – only a look up of positional genomic overlap between the identified sex-DM and sex-DE associations was performed. 3) sex-mQTLs were also not assessed by Czamara et al by modelling sex*SNP interaction terms but were assessed on the condition that a sex-mQTL was identified on the combined-sex analysis followed by sex-stratified analysis. This approach precludes detection of opposite sex-effect QTLs and yields results biased towards QTLs with consistent effect direction while limiting sex-specific QTLs. Therefore, our study stands out in comprehensive integrative analyses of all three omics datasets (sex-DE, sex-DM, sex-eQTL, sex-mQTL) and the approach we implemented brought to light sex-eQTLs and sex-mQTLs with sex-specific effects and that the sex-DMs are enriched for CpGs which are associated with gene expression in placenta.

2. Although the findings reported in this manuscript are interesting, overall the paper remains descriptive, being heavily based on correlations. As per the famous phrase “correlation does not imply causation”, I think we should attempt, whenever possible, to use ways to experimentally assess interesting data-driven hypotheses. We have now the ability to use human trophoblast cell culture with several lines being commercially available and having the sex-of-origin reported (e.g. BTS5, CT27 and CT30 are trophoblast stem cell lines of female origin and BT11 and CT29 trophoblast stem cell lines of male origin). As the authors acknowledge in the discussion, placenta cell composition is complex and the methods that assess cell composition in silico have considerable limitations. Although cell culture systems have their own limitations, the use of male and female human trophoblast cell lines would allow to independently validate sex-DM and sex-DE in a more homogeneous cell population. In addition, cell culture systems would create the opportunity to manipulate DNA methylation either at a global level (e.g. using inhibitors of DNMT, such as zebularine) or at specific loci, such as at those presented in Figure 3.

Response: We agree with the reviewer that functional studies are important for an in-

depth mechanistic interpretation of these findings. We plan to pursue this in the future, but that is beyond the scope of this paper.

3. When reporting their findings regarding sex-mQTLs, the authors refer to some examples that are illustrated in Figure 4. However, in the panels of these figures we only see the location of the genetic variant (rs), but not that of the CpG (cg) or the gene that the text refers to. I think these panels should provide a better visual representation to match it with the text. A similar issue exists for the examples of sex-eQTLs illustrated in Figure 5: we see the position of the genetic variant (rs), but not that of the gene itself.

Response: We have updated both figures (now Figure 5 and 6) showing the CpG and gene relative physical locations on the genome.

4. Related to the above point, when presenting sex-mQTLs, the authors report that the average distance between the genetic variant (rs) and the CpG (cg) is around 390 kb. However, there is no additional analysis or even a discussion on how a genetic variant could induce a sex-DM over such a long distance: could it be chromatin looping between the two loci or another mechanism? Regarding the first scenario, a good point of call would be the recent study by Varberg et al. 2023 (PMID: 37563143), which reported long-range chromatin interactions (by Hi-C) in the human trophoblast cell lines CT27 (female) and CT29 (male) in both stem cell status and upon differentiation to EVT (extravillous trophoblast). I think that exploring and integrating these datasets would give additional strength to the current study.

Response: Thank you for this insightful suggestion. The following text has been added (lines 153-160):

“Across the 1839 sex-mQTLs, the genetic variant’s distance from its associated CpG was inversely correlated with its sex-biased effect on methylation ($r_s = -0.20$, $P = 7.32 \times 10^{-19}$). Genetic variants may induce epigenetic regulation in placenta through long-range chromatin interactions³⁵ and distal transcript regulatory elements such as enhancers.^{41,42} Among our sex-mQTL SNPs, 37 SNPs overlap with predicted placental enhancers by Owen *et al*⁴², and 36 SNPs overlap with predicted placental enhancers by Zhang *et al*⁴¹. The majority of SNPs that showed enhancer overlaps (i.e., 94.5% (35/37) in⁴² and 66.7% (24/36) in⁴¹) were more than 100kb away from the sex-mQTL CpG (Supplementary Table 4).”

We have also integrated our loci showing sex-DM CpG gene expression correlations with Varberg et al’s (PMID: 37563143) chromatin interaction loop in placental trophoblast stem state and EVT cells, and added the following sentences (lines 129-132, 132-134, 139-143):

“Previously, male-bias has been found in a 225 kb locus of differentially methylated regions (DMRs) within the gene body of *CSMD1* and in transcript abundance of *CSMD1*

in placenta³⁴, although the relationship of the DMRs with *CSMD1* transcript abundance was not elucidated.”

“The two CpGs that we found to be correlated with *ZNF300* and *CSMD1* overlap with predicted active TSS-promoter and enhancer peaks, respectively based on ENCODE and RoadMap Epigenomic annotations.”

“The CpG linked to *PSMA8* overlaps with a predicted enhancer and with a long-range chromatin interaction loop (based on high throughput chromosome conformation capture, Hi-C) in placental extravillous trophoblast (EVT) cells.³⁵ The CpG linked to *CADM2* overlaps with a predicted enhancer and active chromatin in EVT cells³⁵ and overlaps with a predicted polycomb-associated heterochromatin by ENCODE and RoadMap.”

5. I understand the need to highlight some poster child examples in the abstract. However, I question the choice of *KCNQ1* and *IGF2R*, for which the authors report in the abstract that their female-biased sex-DM “correlated with increased placental weight”. However, from the results section, we learned that the correlation scores are very weak and for *IGF2R* the associated P value is not even significant if P<0.05 threshold is upheld (*KCNQ1*: cg13536051: rs=0.11, P=0.048 and cg15782852: rs=0.18, P=0.002; *IGF2R*: cg03634777: rs=0.11, P=0.064). There are probably several other specific examples in the study that could be highlighted instead.

Response: Thank you. We have updated the sentence as “We found enrichment of imprinted genes in sex-differentiated placental methylation, including female-biased methylation within the well-known *KCNQ1OT1/CDKN1C* imprinting cluster of maternally expressed genes.”. Note that we have revised the imprinting discussion based on reviewer #3’s suggestions.

Reviewer #3 (Remarks to the Author):

The authors used set out to assess differences in DNA methylation, gene expression and methylation- and expression-QTLs according to the sex of the human term placenta. Samples were collected as part of the study and underwent SNP genotyping with the HumanOmni2.5 Beadchip, DNA methylation profiling with the Infinium Methylation450 Beadchip, with a subset also undergoing RNA-seq. This study is important, since sex is so often overlooked in epigenetic analyses. In particular, male fetuses often cope much more poorly with pregnancy complications, the cause of which is unknown. Further, differences between the sexes are present across the lifecourse, but which are dictated by life in utero is again unknown.

Response: Thank you for recognizing the importance of fetal sex in placenta and pregnancy complications and the added value of our work. We are very grateful to the reviewer for the constructive critique that helped improve the interpretation of the findings.

The paper is well written and clear, though I have several major comments regarding the gene expression correlation with sex-DM, and the section on genomic imprinting.

Response: We appreciate this feedback of the reviewer, and we have made updates and included additional analyses and explanations – which greatly improved the manuscript.

Major point 1: Correlation between sex-DM CpG methylation and gene expression –
Line 99 – 119

I find this section to be underwhelming in its depth of thought and analysis. It is well established that the function of DNA methylation varies with context, whereby methylation close to the TSS impedes transcription, but in the gene body it is positively associated with transcription. As such, the text referring to whether the relationship between DNA methylation and gene expression is ‘positive’ or ‘negative’ is rather clunky, and would be improved with more information on precisely where the methylation in question is. Firstly, I suggest some additions to Figure 3 so that the spatial relationship between sex-DM CpGs and the genes (and their structure) suggested to be impacted by differential methylation can be seen visually. Currently, the location of methylation - ‘body’/‘TSS’ is indicated only in a Supplementary Table.

Response: Thank you for this suggestion that gave us an opportunity to clarify the findings better. We have now specified the locations of the CpGs relative to the genes and provided more context considering prior evidence (which is also suggested by reviewer #1). We have updated the figure, to include CpG-gene spatial relations and UCSC Genome Browser regulatory annotation tracks and placenta chorion-related regulatory tracks to each plot. To effect this, figure 3 is now split into two figures (Figure 3 showing male-hypermethylated CpG-gene correlations, and Figure 4 showing female-

hypermethylated CpG-gene correlations). Regulatory annotations have also been added to supplementary Tables 2 and 3.

We have added the following sentences (lines 112-121).

“Positive sex-DM CpG-gene correlations were more common than negative correlations. Moreover, the majority of CpGs positively correlated with gene expression were in the gene body, whereas the majority of those with negative correlation were in the gene transcription start site (TSS) (**Supplementary Table 3**). These findings align with studies that showed that methylation within gene body is often positively correlated with gene expression while methylation in TSS is often negatively correlated with gene expression.^{30,31} Positive correlation between methylation and gene expression may be explained by mechanisms such as preferential binding of transcription factors to highly methylated sites³², and binding of gene-repressive transcription factors to unmethylated sequences.³³”

Related to this, I could not find any information regarding the genomic distance of the differential methylation to the genes in question, only that they are considered to be ‘nearby’. The only way to check, (unless the CpG is in the TSS, in which case this information is in Supp. Table 3, though I suggest moving to the main text for the highlighted genes) is for the reader to themselves input the CpG accession into the UCSC genome browser. Further, there is no information on other potentially relevant information on the CpGs, such as their position within predicted enhancers or insulators (though these elements can be very tissue-specific, this information remains useful). This could be included in the main text at least for the 8 genes highlighted in Figure 3. For instance, for the first CpG on the list of Supp. Table 3, (cg00874873), this region is predicted in USCS to be an enhancer, and the array probe is just slightly downstream of TFAP2 transcription factor binding sites, which are key in the placenta. A smaller point, but the ‘strong’ / ‘weak’ effect of differential methylation should also be put in the context of distance – i.e. distal differential methylation may have a weaker effect than the same methylation change close-by.

Response: This has been addressed in the revised figures (figure 4 and 5) and within the results text (as mentioned above). We have added distance of CpGs to gene start and TSS and regulatory annotations of the CpGs in Supplementary Table 3.

The text has also been updated with information on relevance of the CpG positions and predicted functions (lines 122-147) as follows:

“The top four strongest male-hypermethylated CpG-gene and top four strongest female-hypermethylated CpG-gene correlations are presented in **Figure 3, A-D** and **Figure 4, A-D**, respectively. Among male-hypermethylated sex-DM CpGs correlated with gene expression, methylation of cg11291313 located in *ZNF300* TSS and *ZNF300* expression showed the strongest inverse correlation (Spearman r (r_s)=-0.56, $P=3.51 \times 10^{-7}$); and cg02563011 located within *CSMD1* gene body (167.4 kb from gene start) and *CSMD1* showed the strongest positive correlation ($r_s= 0.73$, $P=3.44 \times 10^{-13}$).

Previously, male-bias has been found in a 225 kb locus of differentially methylated regions (DMRs) within the gene body of *CSMD1* and in transcript abundance of *CSMD1* in placenta³⁴, although the relationship of the DMRs with *CSMD1* transcript abundance was not elucidated. The two CpGs that we found to be correlated with *ZNF300* and *CSMD1* overlap with predicted active TSS-promoter and enhancer peaks, respectively based on ENCODE and RoadMap Epigenomic annotations. Among female-hypermethylated sex-DM CpGs correlated with gene expression, cg01070760 located in *PSMA8* exon (193 bp from gene start) and *PSMA8* showed the strongest inverse correlation ($r_s=-0.69$, $P=3.07 \times 10^{-11}$); and cg25948255 located within the gene body of *CADM2* (504 kb from gene start) and *CADM2* showed the strongest positive correlation ($r_s=0.65$, $P=1.17 \times 10^{-9}$). The CpG linked to *PSMA8* overlaps with a predicted enhancer and with a long-range chromatin interaction loop (based on high throughput chromosome conformation capture, Hi-C) in placental extravillous trophoblast (EVT) cells.³⁵ The CpG linked to *CADM2* overlaps with a predicted enhancer and active chromatin in EVT cells³⁵ and overlaps with a predicted polycomb-associated heterochromatin by ENCODE and RoadMap. *ZNF300*, *CSMD1*, and *CADM2* have been implicated in cancer cell proliferation, migration, and invasion²⁹⁻³¹, which are features known to be mirrored by the invasive, immunologic and angiogenic properties of placental cells.³² *ZNF300* methylation has been correlated with placental morphology, and invasive placental trophoblast cells may drive sex differences at *ZNF300* methylation.²⁶ ”

We note that distance of a CpG from gene for the 63 CpG-gene pairs was not correlated with the strength of correlation of CpG methylation with gene expression ($r=-0.08$, $p=0.52$).

Major Point 2

The methods section indicates that only the closest gene to the CpG was considered: Line 435 “To evaluate relations between sex-DM CpGs and gene expression, we tested whether methylation at each sex-DM CpG is correlated with placental expression of its closest gene.”

What about other nearby genes? Could you not repeat the analysis and look at genes within a defined window (100 kb?), this would then tell you whether the effect of differential methylation varies by distance. Methylation in a given sex-altered region may be having wide-ranging effects on gene expression, but this is missed in this analysis, and may explain why correlations between differential methylation and differential gene expression are low.

Response: We expanded our analysis to +/- 200 kb distance between CpG and gene start, finding 13 additional correlations after adjusting the p-values for the number of tests performed. The findings did not significantly expand the number of correlations between gene expression and methylation, and largely overlap with the reported correlations based on nearby genes annotations from the UCSC Genome Reference which cover up to 200 kb distance. The Methods and Results have been updated as follows (lines 111-112 and 558-560, respectively). The additional correlations identified have been included in Supplementary Table 3.

Methods (line 558-560): “In further exploration, we assessed correlation between methylation at each sex-DM CpG and placental expression of all genes within 200 kb distance from the CpG site.”

Results (line 111-112): “An additional 13 CpG-gene correlations were identified by testing the correlation of sex-DM CpGs against all genes within 200 kb distance.”

Potentially major point – though perhaps a wording error:

Line 104 – “Methylation at sex-DM CpGs was more likely to be significantly correlated with nearby gene expression in placenta at FDR-adjusted $P < 0.05$ than non-sex-DM CpGs located >500 kb away on the array (63/3429 vs. 395/59411; $\chi^2=61.6$, $P=4.21 \times 10^{-15}$).”

I’m confused by the wording here, else the premise for this comparison is flawed – are you saying that methylation at sex-DM CpGs was more likely to be correlated with nearby gene expression than non-sex DM CpGs (reasonable) or compared to non-sex DM CpGs that are also >500 kb away – this latter option is not a good comparison since you are not separating out the fact of their sex-DM or non-sex-DM status from the distance!

Response: We have now changed the comparison to “non-sex-DM CpGs” as suggested. The results remain consistent. The text is now updated as follows (line 108-111):

“Methylation at sex-DM CpGs was more likely to be significantly correlated with nearby gene expression in placenta at FDR-adjusted $P < 0.05$ than non-sex-DM CpGs on the array (63/3636 vs. 1288/277343; $\chi^2=120.6$, $P=4.67 \times 10^{-28}$).”

Minor points:

Line 110 – you need to cite here PMID: 29376485 Gong et al 2018 (currently reference 19 in your manuscript) since they already found the sex-specific placental DMR at CSMD1 and characterised it extensively.

Response: Thank you. We have added a sentence citing the paper (line 129-132). “Previously, male-bias has been found in a 225 kb locus of differentially methylated regions (DMRs) within the gene body of *CSMD1* and in transcript abundance of *CSMD1* in placenta³⁴, although the relationship of the DMRs with *CSMD1* transcript abundance was not elucidated.”

Whether sex-differentiated placental methylation is linked with gene expression regulation is unclear because no previous study has integrated the two molecular traits – this is not entirely true – your reference PMID: 28234023 a study on sexual dimorphism do integrate DNAm and expression, though they use gene expression data from a previous published study, and again PMID: 29376485 also link placenta

methylation and gene expression

Response: We have updated the sentence (line 63-64), removing "...because no previous study has integrated the two molecular traits". We note that whether sex-DM CpGs are associated with gene expression in placenta has not been tested by the previous studies, although overlap between sex-DM and sex-DEGs at the annotated genes was performed. In our study we assessed if the sex-biased CpGs are involved in gene regulation in addition to a look-up of overlaps between sex-DE and sex-DM.

Line 137 – "7.9% (134/1701) of sex-mQTLs had concordant effects in males and females", ok so what makes them sex-mQTL if they are concordant in males and females? Is it the level of DNAm? Please explain in the text.

Response: Yes, consistent effect denotes directionally consistent effects that differ in magnitude between males and females. We have added the following phrase in the text (line 175), in addition to the definitions provided in the Methods section. "... directionally consistent effects that differ in magnitude...".

Enrichment for Imprinted Genes, Line 228 onwards. The descriptions of imprinting are, in general, imprecise.

Response: Thank you so much for your feedback and the related comments below. We have re-worded the Results section on imprinting and the associated text in the Discussion. The results section now reads as follows (line 307-339):

"The identified imprinting overlaps included a cluster of 13 female-hypermethylated sex-DM CpGs within the well-known *KCNQ1OT1/CDKN1C* imprinting domain in chromosome 11p15.5 genomic region. In human placenta, the *KCNQ1OT1/CDKN1C* imprinting domain harbors maternally expressed imprinted genes such as *CDKN1C*, *PHLDA2*, and *SLC22A18*, a paternally expressed long non-coding RNA that represses imprinted genes in the domain (*KCNQ1OT1/kvDMR1*), and other genes such as *KCNQ1* that exhibit biallelic expression.⁷³ The imprinting domain is regulated by a maternally methylated DMR imprinting control region (ICR) located in intron 10 of the *KCNQ1* gene.⁷³ Out of the 13 sex-DM CpGs, 8 were located within *KCNQ1* at 13.7-165.1 kb distance from the ICR and overlap with histone 3 lysine 27 trimethylation (H3K27me3), a posttranslational epigenetic modification associated with transcriptional repression. Given these annotations, we assessed whether methylation at the 8 CpGs has a potential sex-dependent relationship with placental expression of genes in the chr11p15.5 imprinting domain. Higher CpG methylation showed correlation with increased *KCNQ1OT1* expression in females and trended with decreased *KCNQ1OT1* expression in males (cg03030994-*KCNQ1OT1*: $r_m=-0.26$, $P_m=0.12$; $r_f=0.36$, $P_f=0.04$), increased *CDKN1C* expression in females (cg00446023-*CDKN1C*: $r_m=-0.09$, $P_m=0.58$,

$r_f=0.35$, $P_f=0.046$), and decreased *SLC22A18* and *KCNQ1* expression in males (cg05457684-*SLC22A18*: $r_m=-0.44$, $P_m=0.006$, $r_f=0.04$, $P_f=0.83$; cg05457684-*KCNQ1*: $r_m=-0.46$, $P_m=0.004$; $r_f=0.05$, $P_f=0.79$; cg06960356-*KCNQ1*: $r_m=-0.42$, $P_m=0.009$; $r_f=0.14$, $P_f=0.45$). Imprinting dysregulation of genes in the *KCNQ1OT1/CDKN1C* domain has been linked to Beckwith-Wiedemann syndrome, a disorder of growth regulation characterized by somatic overgrowth and tumor predisposition.⁷⁴ Moreover, increased placental expression of the maternally expressed *CDKN1C*, *PHLDA2*, and *SLC22A18* genes has been linked with fetal growth restriction and smaller neonatal size.⁷⁵⁻⁷⁹ We found that three female hypermethylated sex-DM CpGs showed correlation, albeit weakly, with lower birthweight (cg25548316: $r_s=-0.13$, $P=0.03$), higher placental weight (cg13536051: $r_s=0.11$, $P=0.048$; cg15782852: $r_s=0.18$, $P=0.002$), and higher placenta-birthweight ratio (cg13536051: $r_s=0.2$, $P=0.001$; cg15782852: $r_s=0.14$, $P=0.02$).

Namely, whilst IGF2R is a powerful regulator of placental growth, evidence of its imprinting in human is weak, whereby it is polymorphic/rare. As such, the use of IGF2R by the authors as an exemplar imprinted gene in humans involved in the conflict hypothesis is not appropriate. Further, IGF2R is not involved in Beckwith Wiedemann Syndrome.

Response: Thank you – this has been amended in the revised paragraph (see above).

Line 237 – “such as insulin-like growth factor type 2 receptor (IGF2R) and potassium voltage-gated channel subfamily Q member 1 (*KCNQ1*), which are paternally imprinted (maternally expressed) in placenta and involved in regulation of fetoplacental growth and development ref52, 53”

Reference 52 is a purely mouse paper, with no analysis of IGF2R in humans. Human IGF2R imprinting, whilst it is maternally expressed, is highly polymorphic in and reported in only a minority of individuals (PMID: 16614068). Loss of imprinting of this gene has also not been linked with any human pathology. There is also no strong evidence that the *KCNQ1* gene is involved in fetoplacental growth/development, except of the heart (the wider locus notwithstanding)

Response: Thank you – this has been amended in the revised paragraph (see above).

Line 322, regarding sex biased CpG methylation “This observation is aligned to the role of maternally expressed imprinted genes, as opposed to paternally expressed ones, in limiting fetal growth to conserve maternal resources”

And:

Line 329 “Our observation that epigenetic processes at *KCNQ1* and IGF2R that promote fetoplacental growth were female-biased is not compatible with the well-known male-bias in fetoplacental growth, and raises a potential crosstalk between sex and imprinting regulation in placenta. The male placenta functions near its maximum capacity prioritizing fetal growth, whereas the female placenta prioritizes maternal reserve”

Aside from the fact that, as discussed above, *KCNQ1* and IGF2R are not, in humans, strong candidates for exemplar ‘conflict hypothesis’ genes, I believe the rationale here is

not correct. I do not agree that the male placenta prioritises fetal growth and the female placenta prioritises maternal reserve. All placentas are selfish. Instead, females, differently to males, reduce growth to preserve other fetal reserves in case of maternal stress (such as asthma or pre-eclampsia) whereas males increase growth to try and improve their chances. Different strategies, but both designed to improve fetal chances of survival, regardless of maternal reserve.

Response: Thank you – we agree that the “conflict hypothesis” may not be as strongly conserved in humans, and the associations detected may as well be interpreted in line of other hypothesis. This has been updated in the revised paragraph (see above).

Line 324 “repressed expression of KCNQ1 and IGF2R cause Beckwith-Wiedemann syndrome”

There is no evidence that dysregulated imprinting of IGF2R causes BWS (do the authors mean IGF2?). Further, whilst the KvDMR is highly implicated in BWS pathogenesis, the KCNQ1 gene itself is not – CDKN1C is a much more important gene here. Did you check the expression of other genes in the locus?

Response: We have now assessed whether the CpGs in the *KCNQ1OT1/CDKN1C* imprinting domain are associated with imprinted genes across the long range of the region in human placenta and presented the findings. Please revised paragraph above.

Line 333: “We posit that there may be sex-dependent epigenetic regulation of imprinting at KCNQ1 and IGF2R as a compensatory adaptive mechanism to counter-balance resource allocations to the fetus to prevent undergrowth in females and overgrowth in males”. This is unlikely - there are no sex biases in the clinical disorders caused by LOI at KCNQ1, and by necessity the germline KvDMR imprint is very stable.

Response: Thank you – this has been amended in the revised paragraph (see above).

Minor Comments:

Line 347: “Third, sex is correlated with numerous environmental factors, which makes it difficult to dissociate differences due to biological sex and other factors correlated with biological sex”. I entirely disagree here – a unique strength of this study is that you are looking at the placenta, and therefore at sex differences in utero. This is prior to the manifestation of the vast environmental differences faced by ex-utero males and females. Prenatally, the environment the mother provides for her male or female fetus, aside from innate fetal hormonal and developmental differences (which I do not believe the authors mean when they say ‘environmental’), is likely to be very similar regardless of sex, and so in fact not subject to environmental differences.

Response: The sentence is now taken out.

Line 150: “None of the 14 genes exhibited sex difference in non-placental tissues from the genotype tissue expression (GTEx) portal, aligning with evidence of poor correlation of sex-differentiated autosomal gene expression between term placenta and 42 non-reproductive adult tissues from GTEx”

-This is not surprising since the cited report finds sex specific differences overall from GTEx to be very tissue specific (as the authors note on line 164). However, it is not clear whether the placenta has a uniquely poor correlation with other tissues? (i.e. compared to say, Liver vs Brain or Lung?) – it would be interesting to expand slightly on this with more computational analysis, to establish whether placenta is an outlier (as is often the case in terms of tissue specific gene regulatory profiles.)

Response: This is an important point, and a recent study in the POPS cohort has identified 71 placenta-enriched transcripts compared to 49 tissues in GTEx (PMID: 33976128; <https://www.obgyn.cam.ac.uk/placentome/>), making the placenta the 4th highly ranked tissue with the largest number of tissue-specific transcripts. Tissue-specific scores of RNA transcripts using the tau score (a tissue-specificity measure) are also available in the Human Protein Atlas (proteintlas.org). The sex-DE genes we identified, however, are not among the list of placenta-specific RNA transcripts in the POPS study nor in the Human Protein Atlas. We have added the tau scores and tissue specificity profiles of the genes to Supplementary Table 6. We have also added the following text in the results section (line 198-200):

“None of the sex-DE has placenta-specific RNA expression as compared to other human tissues based on tissue-specificity metrics in the Human Protein Atlas or in a placental RNA-sequencing study (**Supplementary Table 6**).^{48,49}”

Line 239, plus abstract and various other places: Please don't use the term imprinted to mean the silencing of an imprinted gene - say silenced, or repressed allele.

Response: Thank you, the language has been updated.

Regarding the discussion of the overrepresentation of imprinted genes in sex-biases – rather than focusing on resource allocation– perhaps the authors could speculate instead about whether particular epigenetic facets of imprinted domains could leave them vulnerable to sex-specific alterations?

Response: This is an interesting point, and we have incorporated it in the revised manuscript Discussion (lines 417-434) as follows:

“We found a previously unrecognized enrichment of imprinting genes in sex-differentiated placental methylation, including a cluster of female-biased methylation

within the maternally expressed *KCNQ1OT1/CDKN1C* imprinting domain implicated in Beckwith-Wiedemann syndrome.^{73,74} In our data, higher methylation of the female-hypermethylated CpGs was associated with female-specific upregulation or male-specific downregulation of maternally expressed genes in this imprinting domain. These findings suggest a potential crosstalk between sex and imprinting regulation in placenta. The unique epigenetic architecture of imprinting in human placenta may be permissive to epigenetic alterations. For example, in the *KCNQ1OT1/CDKN1C* cluster, loss of imprinting is variable and gene repression by the *KCNQ1OT1* lncRNA is incomplete.⁹⁰ Human placental DMRs are uniquely highly polymorphic.^{69,73,91} Placental DMRs are associated with variable histone marks⁹² which are mechanistically linked to DNA methylation that influences imprinting stability⁹³ and potentially with underlying genetic variation.⁹¹ Collectively, these studies and our findings suggest that the imprinting domains may be vulnerable to epigenetic alterations from other factors that influence allelic bias in a sex-dependent manner, but future work should investigate this and its relevance to sex differences in pregnancy physiology and clinical complications.”

Line 269: Enrichment for TFs such as hormone receptors - perhaps this is not a surprising finding in the context of sex differences, NR2C2 is involved in spermatogenesis – could the authors comment further on this?

Response: We have added the following sentence (lines 366-370): “*NR2C2* (nuclear receptor subfamily 2 group C member 2) encodes the testicular receptor 4 protein TR4. In vivo mice studies have shown that *nr2c2* plays key roles in fetal growth, early postnatal survival, fertility, and sensitivity to environmental stimuli.^{82,83} Male *nr2c2*-null mice exhibit delayed spermatogenesis and reduced fertility⁸³, and females exhibit maternal behavioral abnormalities.⁸²”

Line 282: “methylation sites that were sex-biased were not enriched for TF binding’ this should say ‘predicted’ TF binding, or instead ‘TF binding sites’.

Response: This is now updated (line 382): “...enriched for predicted TF binding”.

The Supplementary table PDF is not complete and appears to only have table 1

Response: The pdf converted table which may have been incomplete based on the way the online submission system converts files to PDF. However, there is an additional Excel spreadsheet that includes all supplementary tables.

RESPONSES TO REVIEWERS' COMMENTS

Reviewer #1 (Remarks to the Author):

In this revised version, the authors have addressed my review questions, added new analyses, and I have no further comments. Just one side note: Ref 68 has been published in Nature Genetics, 55:1807–1819 (2023), and this reference also showed a maternal-only association between a variant in intron 10 of KCNQ1 and placenta weight.

We have now replaced the preprint citation by the published paper citation. We have also added the following sentence about the *KCNQ1* variant's maternal parent-of-origin effect on placenta weight (line 315-316): "A genetic variant in intron 10 of KCNQ1 has shown a maternal parent-of-origin effect on placental weight.⁷²"

Reviewer #2 (Remarks to the Author):

In the revised manuscript, Tekola-Ayele et al. addressed many of the points raised on the original manuscript by myself and my colleagues who reviewed the paper. Although they did not perform any additional experimental work to address the functional relevance of their findings (point 2 in my previous comments), the analyses performed are welcome additions to the paper. On the flip side of the coin, although there are some differences from the work performed by Czamara et al. (PMID: 38600394) regarding the approaches used to analyse their data (these differences are mentioned in the reply to reviewer's comments), in my view the main messages of the two papers are highly similar:

- sex has a strong impact on placental DNA methylation, with most sex-DM being hypermethylated in males;
- differences in placental gene expression between the two sexes are limited;
- there are very few loci for which there is a direct triad relationship methylation-expression-phenotype.

The following sentence has been added to the Discussion section (paragraph 2, line 397-403). "The largely distinct sex-DM and sex-DE associations detected in the present study align with a recent report.²⁰ Our study implemented more omics-integrated analyses including directly testing association of sex-DM CpGs with nearby gene expression, identifying sex-eQTL associations, and implementing analyses models that included interaction terms between sex and genetic variants. These approaches yielded unique insights that sex-DM CpGs were more likely to be associated with nearby gene expression alteration and enabled detection of several previously undetected sex-mQTLs and sex-eQTLs with single sex effect."

The example on the imprinted cluster located on chromosome 11p15.5 remains rather underwhelming, because most of the correlations listed are weak. There is no explanation for why this cluster of imprinted genes may be more "susceptible" to acquiring sex-DM compared to other imprinted loci.

In the results section, we inserted "albeit weakly...." to note that the correlations with phenotypes are not strong (line 333), and discussion of the corresponding CpG-trait correlations was taken out. In addition, the following statements in the Discussion section provide explanation for why the *KCNQ1OT1/CDKN1C* cluster may potentially be "susceptible" to sex-DM, with a caveat that future studies will be needed (line 425-433): "For example, in the *KCNQ1OT1/CDKN1C* cluster, loss of imprinting is variable and gene repression by the *KCNQ1OT1* lncRNA is incomplete.⁹² Collectively, these studies and our findings suggest that the imprinting domains may be vulnerable to epigenetic

alterations from other factors that influence allelic bias in a sex-dependent manner, but future work should investigate this and its relevance to sex differences in pregnancy physiology and clinical complications.”

As a minor point, the authors should check their list of references. After the edits made, some papers do not seem to be cited any longer in the text.

We have now updated the references and citations.

Reviewer #3 (Remarks to the Author):

I find the manuscript to be much improved and the authors carefully answered all my previous points. My remaining comments below are largely stylistic, though there are still a couple of areas requiring a bit more precision regarding descriptions of imprinting. I think a couple of edits will have been lost to formatting - please also see pdf file attached here.

Abstract:

Line 33: “...known KCNQ1OT1/CDKN1C imprinting cluster of maternally expressed genes” – they are not all maternally expressed. The ncRNA (and locus-controlling transcript) KCNQ1OT1 is paternally expressed.

Corrected.

Line 61 – nearby genomic (no s) region

Corrected.

Line 102: “Differentially methylated sites may be involved in transcript regulation depending on context.

Line 103: “Therefore, we assessed whether the sex-differentially methylated sites are correlated with nearby gene expression more than expected by chance.

These are two separate points. The first could be described more clearly – suggest instead “Differentially methylated sites may be involved in transcript regulation, but their precise effect is likely to depend on the genomic context”.

Regarding the second point, whether sex-DMs are correlated with gene expression more than expected by chance, is asking whether there is some selection for sex-DMs to regulate genes. Suggest moving lines 103-108 ‘Therefore we assessed....against all genes within 200kb distance’ down, and addressing your context-dependent analysis first, following line 102 directly with the end of Line 108 – “Positive sex-DM CpG-gene correlations....”

We have expanded the first sentence as suggested and made updates to improve flow (line 102-115).

It is worth clarifying in the text what you mean by positive sex-DM CpG-gene correlations – I suppose it is that the methylated CpG state correlates with increased gene expression?

We have clarified the sentence as suggested (line 112)

Line 112: “These findings align with studies that showed that methylation within gene body is often positively correlated with gene expression while methylation in TSS is often negatively correlated with gene expression”

More of a stylistic issue here but the description used by the authors here “findings align with studies that showed” is not quite appropriate – the relationship between gene-body methylation and transcription has

been well established since 1984 and should be described as such. Something like: “These findings are consistent with the common theory that methylation within the gene body is positively correlated with gene expression and methylation in TSS is negatively correlated with gene expression” (insert key references such as Jones, P. A., 1999 PMID: 10087932, or Jones P.A., 2012 PMID: 22641018, Laurent et al., 2010 PMID: 20133333).

We have updated the sentence and added the suggested citations (line 115-118), which now reads: “These findings are consistent with long known theory and recent evidence that methylation within gene body is often positively correlated with gene expression while methylation in TSS is often negatively correlated with gene expression.³⁰⁻³⁶”

Line 114 “Positive correlation between methylation and gene expression may be explained by mechanisms such as preferential binding of transcription factors to highly methylated sites and binding of gene-repressive transcription factors to unmethylated sequences.” I don’t think this sentence needs to be there, it is a bit misleading and detracts from the main point.

We have deleted the sentence.

I suggest inserting the lines 103-108 ‘Therefore we assessed....against all genes within 200kb distance’ from above into Line 117: “We then assessed whether the sex-differentially methylated sites are correlated with nearby gene expression more than expected by chance. Methylation at sex-DM CpGs was more likely to be significantly correlated with nearby gene expression in placenta at FDR-adjusted $P < 0.05$ than non-sex-DM CpGs on the array (63/3636 vs. 1288/277343; $\chi^2 = 120.6$, $P = 4.67 \times 10^{-28}$). An additional 13 CpG-gene correlations were identified by testing the correlation of sex-DM CpGs against all genes within 200 kb distance.

I think you need to follow this with some explanation - Why did you do this calculation? - What do the authors think the significance of their finding that sex-DMs correlate with gene expression more highly than that expected by chance? (when comparing to all other CpGs on the array). This is not discussed, but is a key point.

We have updated the paragraph and added a sentence to clarify the implication of the finding (line 107-109). “These findings suggest that sex-dependent methylation is not randomly distributed across the genome but is clustered at CpG loci more likely to have impacts on gene transcript levels”

The end of this section is great.

Thank you.

Line 146, small errors/grammar: “...suggesting that genetic variants anywhere within the 1Mb distance can have sex-dependent effects on methylation in placenta.”

We have updated the sentence.

Line 157: Across the 1839 sex-mQTLs, the genetic variant’s distance from its associated CpG was inversely correlated with its sex-biased effect on methylation ($r_s = -0.20$, $P = 7.32 \times 10^{-19}$). Genetic variants may induce epigenetic regulation in placenta through long-range chromatin interactions³⁵

To make this clearer I suggest the following change:

Across the 1839 sex-mQTLs, the genetic variant’s distance from its associated CpG was inversely

correlated with its sex-biased effect on methylation ($r_s = -0.20$, $P = 7.32 \times 10^{-19}$) which was to be expected, since previous evidence indicates that >75 % gene-regulatory-element interactions are short range (Fuentes et al., ELife 2018, PMID: 30070637). However, genetic variants may induce epigenetic regulation in placenta through long-range chromatin interactions³⁵ and distal transcript regulatory elements such as enhancers.^{41,42}

Thank you, we have updated the sentence as suggested.

Line 304 "...genes such as KCNQ1 that exhibit biallelic expression". This is not quite correct – the cited paper demonstrates that KCNQ1 is imprinted in fetal placenta, (and it is shown elsewhere to be imprinted in many other tissues). You could say 'in term placenta', but in this case you need to say that KCNQ1OT1 is also biallelic at term, as Monk et al., show in the cited paper.

We have expanded the sentence to clarify the ambiguity, as suggested (line 309-313): "In human placenta, the KCNQ1OT1/CDKN1C imprinting domain harbors genes such as CDKN1C, PHLDA2, and SLC22A18 that are maternally expressed in first trimester and term placenta, a long non-coding RNA (KCNQ1OT1/kvDMR1) with paternal expression in first trimester and biallelic expression in term placenta, and other genes such as KCNQ1 which exhibits maternal expression in first trimester placenta and biallelic expression in term placenta."⁷⁸

Line 317 – 'imprinting dysregulation...' rather implies that your observations are akin to imprinting being disrupted according to sex, however, I don't think this is the case - KCNQ1OT1 is paternally expressed, whilst CDKN1C is maternally expressed, so a mix of paternally and maternally expressed genes at the same locus exhibit upregulation in the presence of maternal hypermethylation. To me, this indicates that this is unrelated to imprinting, or at least it is not a strong pattern. It is fine to mention imprinting dysregulation and links to disease, but perhaps make clear that this is simply another way that dosage of these genes is altered.

We have revised the sentence to clarify the intended meaning (line 327-330). "The KCNQ1OT1/CDKN1C domain has been linked to diseases and trait differences. For example, imprinting dysregulation of genes in the KCNQ1OT1/CDKN1C domain has been linked to Beckwith-Wiedemann syndrome, a disorder of growth regulation characterized by somatic overgrowth and tumor predisposition."⁷⁹

Also, you mention SLC22A18, but according to Supp Table (and the positions of the Sex-DM CpGs) they are actually in SLC22A18AS.

Thank you. We have corrected the supplementary table, replacing SLC22A18AS by SLC22A18 as indicated in the source databases.

From Supplementary Table 11: 29 of the Sex-DM CpGs are in TP73 – perhaps the authors could comment on this gene, there is a large enrichment here, and also in HOXA4 (19 here). Also, the 8 Sex-DM CpGs in KCNQ1 are listed twice (due to the gene being derived from both otago and geneimprint lists) giving a false impression of the number of CpGs present in this gene.

We added the following sentence in line 302-305: "The largest regional cluster of sex-DM CpGs was a 63 kb region in the maternally expressed TP73 gene and the homeobox family genes

(HOXA3, HOXA4, HOXA5, HOXB2, HOXB3, and HOXC9), known for their crucial roles in regulating placental and fetal development.⁷⁶”

Lines 401-416: I find the summary and potential reasoning for imprinting sex bias to be much improved.

Thank you.

Line 404 - In our data, higher methylation of the female-hypermethylated CpGs was associated with female-specific upregulation or male-specific downregulation of maternally expressed genes in this imprinting domain. These findings suggest a potential crosstalk between sex and imprinting regulation in placenta.”

Related to my query above, and now misstated here in the discussion, the authors mention ‘female-specific/male-specific up/down regulation of maternally expressed genes’ - KCNQ1OT1 is one of the genes whose expression increases in response to increased CpG methylation in maternal placentas, but it is paternally expressed, which again calls into question the rationale regarding sex and ‘imprinting crosstalk’. I feel the following discussion by the authors on unique epigenetic architecture to be a much more persuasive argument towards the apparent enrichment of imprinted genes here.

Thank you, “maternally expressed” was incorrectly mentioned in the sentence “...downregulation of maternally expressed genes in this imprinting domain” as both maternally and paternally expressed genes exhibited male-specific or female-specific up/downregulation, which is the rationale for sex-imprinting crosstalk. The sentence now reads “...downregulation of genes in this imprinting domain” (line 422).